# New Taxonomic Arrangement of *Dicranella* s.l. and *Aongstroemia* s.l. (Dicranidae, Bryophyta)

**DOI:** 10.3390/plants12061360

**Published:** 2023-03-17

**Authors:** Vladimir Fedosov, Alina Fedorova, Elena Ignatova, Jan Kučera

**Affiliations:** 1Biological Faculty, Lomonosov Moscow State University, 119234 Moscow, Russia; 2Botanical Garden-Institute, FEB RAS, 690024 Vladivostok, Russia; 3Main Botanical Garden, Russian Academy of Sciences, 127276 Moscow, Russia; 4Department of Botany, Faculty of Science, University of South Bohemia, 370 05 České Budějovice, Czech Republic

**Keywords:** Haplolepidous mosses, *nad*5 G1 intron, phylogenetics, polyphyly, morphological convergence, *trn*S-*trn*F region, *trnK*, biodiversity

## Abstract

The recent molecular phylogenetic study of the families Aongstroemiaceae and Dicranellaceae, which resolved the genera *Aongstroemia* and *Dicranella* as polyphyletic, indicated the need for changes in their circumscription and provided new morphological evidence to support the formal description of newly recognized lineages. Following up on these results, the present study adds another molecular marker, the highly informative *trnK–psbA* region, to a subset of previously analyzed taxa and presents molecular data from newly analyzed austral representatives of *Dicranella* and collections of *Dicranella*-like plants from North Asia. The molecular data are linked with morphological traits, particularly the leaf shape, tuber morphology, and capsule and peristome characters. Based on this multi-proxy evidence, we propose three new families (Dicranellopsidaceae, Rhizogemmaceae, and Ruficaulaceae) and six new genera (*Bryopalisotia, Calcidicranella, Dicranellopsis*, *Protoaongstroemia*, *Rhizogemma,* and *Ruficaulis*) to accommodate the described species according to the revealed phylogenetic affinities. Additionally, we amend the circumscriptions of the families Aongstroemiaceae and Dicranellaceae, as well as the genera *Aongstroemia* and *Dicranella*. In addition to the monotypic *Protoaongstroemia* that contains the newly described dicranelloid plant with a 2–3-layered distal leaf portion from Pacific Russia, *P. sachalinensis*, *Dicranella thermalis* is described for a *D. heteromalla*-like plant from the same region. Fourteen new combinations, including one new status change, are proposed.

## 1. Introduction

Molecular phylogenetic studies of bryophytes [1,2,3,4,5,6,7,8,9,10] have shown that the traditional morphology-based circumscriptions of genera are often biased by morphological convergence and in reality comprise a suite of phylogenetically unrelated lineages. This is especially the case for genera with reduced morphology, such as pioneer species with short life cycles, e.g., *Entosthodon* Schwägr. ex Hornsch., *Physcomitrium* (Brid.) Brid. [11,12], or *Ditrichum* Timm ex Hampe [8], although the larger pleurocarpous mosses, such as in the traditionally delimited *Hygrohypnum* or *Hypnum*, have also been shown to be prone to the homoplasic retention of distinct morphological features in unrelated lineages [7,13,14,15].

It was thus perhaps not very surprising that a molecular-phylogenetic study of the northern temperate genera of *Aongstroemia* Bruch & Schimp. and *Dicranella* (Müll. Hal.) Schimp. [10] revealed a striking polyphyly in the existing delimitation of these genera. The traditional morphological circumscription of the genus *Dicranella* included plants of small size with a stem central strand, elongated linear-lanceolate to subulate leaves, costae with guide cells in the cross-section, predominantly dioicous sexual condition, and dicranoid peristome [16,17,18,19,20]. This allowed for a considerable variability of the other gametophytic and sporophytic traits, which found a reflection in the molecular differentiation of the thirteen analyzed species of the genus that were found in seven different phylogenetic lineages of haplolepidous mosses (the subclass of Dicranidae), three of which could be considered orphaned in the system of the representatives of dicranids analyzed to date. On the other hand, the genus *Aongstroemia*, originally introduced for a single species, *A. longipes*, was soon substantially expanded [16] to include most species of the modern *Dicranella* s.lat., and later again reduced [21] to harbor species sharing the julaceous habit originating from the ovate leaves, which are broadly rounded to acute or slightly attenuate at the tips and appressed to the stem. Despite this restriction in the generic concept of *Aongstroemia*, the rate of cryptic molecular diversity was similar to that revealed in *Dicranella*; Bonfim-Santos et al. [10] showed that the three analyzed species (out of the 11 accepted names in the genus) appear in three lineages belonging to three currently recognized families. Although the polyphyly of both genera was demonstrated quite convincingly, this study has not yet resulted in a taxonomic treatment. The major reason for this was the insufficient taxonomic sampling, particularly in *Dicranella*: of the 161 accepted species according to the Tropicos database [22], plus the 47 accepted species in *Leptotrichella* and five accepted names in *Anisothecium*, less than one tenth have been phylogenetically studied, which means that the generic and familial assignment of the bulk of the species remains pending after the splitting of *Dicranella* according to the obtained results. The recent description of a previously unknown dicranelloid moss from SW Portugal, which necessitated the erection of a new genus, *Neodicranella* Porley & Fedosov, following the assessment of molecular affinities [23], confirms that the diversity of dicranelloid mosses has not been fully captured, even in the relatively well-surveyed Europe. Although a thorough taxonomic revision of all included taxa and checking of the type material would be most appropriate, such a revision will hardly be possible in the near future for a complex and species-rich genus such as *Dicranella*, where many of the accepted species are based on a few historical collections from southern tropical countries, whose localities are difficult to access. Nomenclaturally, it would nevertheless be more relevant if the types of the generic names placed earlier in synonymy with *Dicranella* and *Aongstroemia* were designated, and their phylogenetic affinities known; a task that has been partly accomplished and the missing pieces of information do not threaten the stability of most proposed taxonomic solutions. Moreover, further accepting clearly polyphyletic taxa is in our opinion a less desirable alternative than establishing a baseline for further development of a phylogenetically-based system of haplolepidous mosses, which can be further elaborated as soon as new information appears.

Although we generally followed the molecular sampling of Bonfim Santos et al. [10], who employed only organellar markers, plastid *trnL–trnF* and *rps4* and mitochondrial *nad5* intron 2, with respect to the absence of reasonably informative nuclear markers that would be generally used in subclass-spanning phylogenetic studies of mosses, we deepened the molecular sampling to include the highly informative *trnK–psbA* region, which has been used with success, e.g., in the treatments [24] or [25], and also sampled the two variable spacers flanking the gene for tRNA-Thr, which is located between the previously sampled *trnL–trnF* and *rps4*. The purpose of this was the testing of the weakly supported deeper nodes of Dicranidae, which was one of the unresolved questions in [10] that we aimed to address in our novel analyses. Secondly, we broadened the sampling in several critical groups, such as the South American representatives of *Dicranella* assigned to *Anisothecium* by Mitten [26], accessions of *Neodicranella* and several putatively new taxa of unclear affinity from North Asia, which were not included in [10]. We also deliver arguments for treating the two varieties of *Dicranella schreberiana* (Hedw.) Hilf. ex H.A. Crum & L.E. Anderson at the species level and resolve the molecular variation between *D. varia* (Hedw.) Schimp. and *D. howei* Renauld & Cardot. This study thus represents a state-of-the-art taxonomic treatment based on both previously published and newly obtained molecular and morphological data, which is expected to be updated, particularly for the southern and tropical taxa.

## 2. Results

The concatenated matrix consisted of 6381 aligned sites, of which 563 belonged to L partition, 975 to T partition, 728 to R, 2899 to K, and 1216 to N. Indels scored for the L, R, and N partitions yielded an additional 193 binary sites, and 251 indels were scored for the T and 218 for the K partition. The partitions corresponding to the dataset used by Bonfim-Santos et al. [10], i.e., L, R, and N, contained 907 variable and 603 parsimony-informative sites, the T partition had 526 variable and 346 parsimony-informative sites, and the K partition contained 1410 variable and 901 parsimony-informative sites.

The trees inferred from the combined L, R, and N data, and those with an added T region dataset, had essentially identical topology, with generally higher support values from the expanded dataset. The addition of indel data generally further improved the support values, without changes in topology at the supported nodes, however only when T indel data were not considered. At the same time, the trees estimated from the separate analysis of K data, which again had generally higher support node values in the version with SIC-coded indels included, yielded a topological incongruence compared to the trees derived from LTRN data, with respect to the estimated affinities of *Chrysoblastella chilensis*, *Neodicranella hamulosa,* and *Archidium +* Leucobryaceae and Grimmiales clades. Hence, we present here the results representing the total evidence of the fully concatenated dataset (LTRKN) in Figure 1, and the trees resulting from partial analyses of the LTRN and K data are presented as the Appendix A. In the following description of the results, we only comment on results differing from those obtained by Bonfim-Santos et al. [10], which was used as a reference.

In contrast to this study, after reconsidering the reading of several bases in the raw chromatograms, the position of *Pseudoditrichum mirabile* was newly assessed as unresolved among the basal protohaplolepidous clades (*Flexitrichum*, *Scouleria + Drummondia + Hymenoloma* clade) and the clade containing all other analyzed taxa, i.e., *Bryoxiphium +* rest of Dicranales incl. Grimmiales), rather than being found in the basal grade of Dicranales. This position was shared by the analyses resulting from the LTRN and K datasets (Appendix A and Figure 2). Additional *Dicranella staphylina* accessions from northeastern Asia (Putorana, Yakutia, and Khabarovsk Territory) were found in the maximally supported clade with European accessions, although three accessions collected in the heart of the permafrost zone of northeast Asia (BF59, FDt107, and 116) differed in several substitutions, despite the absence of notable morphological differences, except for the slightly more robust habit. European accessions of the previously unanalyzed *D. humilis* from the Czech Republic and Russian Leningrad Province proved identical and distinct from the rest of the analyzed accessions of *D. rufescens* and Far Eastern accessions earlier referred to *D. humilis* based on their seemingly inclined capsules (yet all sporulating collections from that area were collected with immature sporophytes, which prevented the assessment of a basal membrane height). Additionally, the accession FDt119 from plants morphologically approaching *D. humilis* collected in Sakhalin Island was found to be molecularly distinct, in a sister position to the maximally supported *D. rufescens* clade. The topology between the maximally supported *D. rufescens + humilis* clade, *D. crispa + subulata* clade, and the crown clade of Dicranales has not been resolved, even with our deeper molecular sampling, yet the analysis of *trnK* data alone (Appendix A) yielded an unsupported clade containing *D. rufescens + humilis* plus *D. crispa + subulata* lineages. Stronger support (BI PP 1 but without support from ML) was found for the clade containing these two lineages and the clade containing the core Dicranales. The ambiguous affinities of these two lineages might partly result from the ambiguous affinities of the Archidiaceae + Leucobryaceae clade, which was resolved in the sister position to the clade containing Grimmiales based on the LTRN data (now with BS 94/PP 1 support), as opposed to the unsupported (BS < 50/PP 0.79) sister position of the Archidiaceae + Leucobryaceae clade to the rest of the Dicranales, excluding *D. rufescens + humilis* and *D. crispa + subulata* lineages, in the analysis of K data alone. The signal from LTRN data was stronger than that of K data, weakening the support for Archidiaceae/Leucobryaceae plus Grimmiales clade to BS 67/PP 0.94 in the combined LTRKN analysis.

Support for the core Dicranales clade has grown substantially in both ML (BS 85–100 according to the dataset) and BI (PP 1), and similar support was obtained for the sister relationship between *Amphidium* and the rest of the core Dicranales clade. The same is true for the three larger clades within the core Dicranales, the first one including *Schistostega*, Rhabdoweisiaceae, Ditrichaceae, Pottiaceae, Bruchiaceae, and a maximally supported clade containing accessions of *Rhamphidium, Symblepharis krausei,* and *Dicranella vaginata*; the second containing Dicranaceae, Fissidentaceae, *Chrysoblastella*, *Bryowijkia,* and Dicranellaceae; and the third containing Aongstroemiaceae.

The nearly maximally supported (BS 98–100/PP 1) Dicranellaceae clade does not contain northern representatives of *Dicranella* s.l., except for the genus in its amended delimitation itself, and the newly analyzed accession of *Dicranella polii* from Madagascar appeared in a poorly supported clade with two accessions of the genus *Garckea,* which itself appeared nested within a maximally supported clade containing accessions of *Microcampylopus*, *Campylopodium*, and *Leptotrichella flaccidula*. This clade is resolved in sister position to the clade containing *Aongstroemia filiformis* s.lat. accessions. The Madagascan and Reunion accessions of that species appear molecularly distinct from the Neotropical accessions, which earlier were recognized as *A. jamaicensis* Müll. Hal. Within the *Dicranella* s.str. clade, accessions of *D. cerviculata* formed a clade sister to the remaining accessions of the genus. Within the latter, two molecularly identical accessions of *D. heteromalla/Campylopus pyriformis*-like plants from Southern Kuril Islands are separated in a maximally supported clade and are described below as *Dicranella thermalis*. The maximally supported sister clade to the *D. thermalis* clade consists of (1) the nearly maximally supported clade containing accessions of *D. curvipes* and (2) the unsupported clade containing the accessions of *D. heteromalla*. Within this clade, two smaller clades can be recognized, one with high support containing both European and non-European plants referable to this species, and the other unsupported clade containing only accessions from the Russian Far East, which might in the future receive formal status upon a detailed account of their molecular and morphological variability.

Within the now maximally (except for ML BS 87 in the LTRN-based tree) supported Aongstroemiaceae sensu [10], the sister group relationship between *Dicranella varia* s.lat. *+ D. howei + D. pacifica* clade and the rest of the taxa was confirmed, yet the clade containing Aongstroemiaceae without the *D. varia* group now only has weak support (BS 63–78 only from K data and PP 0.67–0.97 according to dataset and indel scoring), with respect to the inclusion of the newly analyzed basalmost lineage containing the plants described below as a new genus, *Protoangstroemia*. Within the *D. varia* group, four maximally or nearly so supported lineages could be recognized. Apart from *D. varia* and *D. howei*, the newly analyzed *D. pacifica* appeared sister to *D. varia + howei* clade, and four accessions, containing the RF42, which earlier was assigned to *D. varia* but now is referred to *Dicranella varia* var. *obtusifolia* Berggren raised to the species rank below, form a lineage sister to the rest of the entire *D. varia* group. Within the Aongstroemiaceae s.str. clade, the basal grade consists of *Protoangstroemia* and the maximally supported lineages of *Diobelonella*, *Dichodontium,* and *Neodicranella*. However, the clade containing accessions of *Neodicranella* appears in a different position in the analysis of LTRN and K data (cf. Appendix A), essentially unresolved in the grade between *Diobelonella* and *Dichodontium* according to LTRN data but deeply nested within *Aongstroemia* s.lat., sister to the *Dicranella grevilleana + Aongstroemia longipes* clade according to K data. The crown clade (*Aongstroemia* s. lat.) contains the *Dicranella grevilleana + Aongstroemia longipes*, *Dicranella schreberiana* var. *robusta*, *D. schreberiana* var. *schreberiana*, *Hygrodicranum bolivianum*, *H. herrerae*, and *Dicranella campylophylla + D. hookeri + Polymerodon andinus* clades. Additional accessions of *Aongstroemia longipes*, *Dicranella grevilleana*, *D. schreberiana* var. *schreberiana*, *D. campylophylla,* and *D. hookeri* support the distinctness of *Aongstroemia longipes* from *Dicranella grevilleana*, *Dicranella schreberiana* var. *robusta* from var. *schreberiana*, and of *D. campylophylla* from *D. hookeri*, yet a more detailed study of taxa in this group needs to be performed in the future with respect to one isolated accession of *Dicranella schreberiana* s.lat. from Russia (RF40), the similarly isolated accession of *D. hookeri* RF65, and the nested position of *D. campylophylla* TJH13 within the clade, which otherwise contained specimens referable to *D. hookeri*.

## 3. Discussion

Our trees are largely congruent with those published by [10,23], yet bring more resolution to the relationships among the haplolepidous lineages, identify affinities of the seven previously unsampled species, and verify the previously assessed affinities using additional accessions of previously insufficiently sampled taxa.

***Dicranella staphylina*.** The totally orphaned position of *Dicranella staphylina* within the system of haplolepidous mosses came as one the most surprising results of the phylogenetic reconstruction by [10]. This moss has to date been known from Europe essentially only from its gametophytic stage, which does not have any distinct autapomorphic traits. Several immature sporophytes have only been observed by [27]. They had yellow to orange seta (speculated to be red at maturity), erect, symmetrical, smooth capsules with irregular and incrassate exothecial cells and few stomata, longitudinally striate peristome teeth bifid to the middle, and unmatured spores 15–20 µm. The character of annulus was not mentioned. Neither of these characters is outstanding among northern *Dicranella* s.lat. species. Unexpectedly, sporulating plants not matching the description of any other known species of the genus were recently discovered in the north Siberian Putorana Plateau. They had yellow setae, asymmetric furrowed capsules, bright red peristome, and well-developed revoluble annulus (Figure 2). Surprisingly, the molecular barcoding of these plants revealed their identity with the previously analyzed accessions of *D. staphylina* and, indeed, the gametophytic characters matched the European material except for plants from Khabarovsk Territory, discovered upon the subsequent herbarium revision, which lack the characteristic rhizoidal tubers. The difference in the capsule shape as compared to [27] might result from the ontogenetic stage, when young straight and smooth capsules may also become curved and furrowed with age. With respect to the isolated position of *Dicranella staphylina*, we propose a new genus and family to accommodate it.

The sister group relationship between *D. staphylina* and the clade containing both the rest of the order Dicranales (including Archidiaceae and Leucobryaceae) and the species currently recognized within the order Grimmiales opens the question of the ordinal placement of the lineage containing *D. staphylina*. While the resolved topology based on LTRN and K data differs in the assessment of affinities of the *Archidium* + Leucobryaceae clade, both topologies agree on the nested position of the currently recognized Grimmiales within Dicranales, should the *D. staphylina*-lineage remain in Dicranales. *D. staphylina* has a fairly typical dicranoid peristome with triangular, in basal and median part longitudinally striolate rather than filiform teeth, split to half of their length into unequal lobes, without a basal membrane (Figure 2), which clearly fits the description of the dicranoid peristome type by [28]. This favors the idea of including this lineage in the delimitation of Dicranales, rather than establishing an isolated new order to accommodate it, suggesting that the dicranoid peristome is the plesiomorphic character state for the whole large lineage, from which the more derived peristomes in Grimmiales and Pottiaceae evolved (see also [9] for a discussion of the secondarily modified peristomes in, e.g., *Glyphomitrium* Brid. or *Pseudoblindia* Fedosov, M. Stech & Ignatov of Rhabdoweisiaceae). The idea of a broad Dicranales, with the currently recognized Grimmiales being lowered to the rank of suborder, has further support from the absence of any derived morphological trait that is typical for the earlier diverging lineages, such as *Catoscopium* Brid., *Distichium* Bruch & Schimp., *Bryoxiphium* Mitt., or *Pseudoditrichum* Steere & Z. Iwats.

***Dicranella rufescens* and *D. humilis*.** These two species share the red color of their stems [29], rather sparsely foliated stems with leaves hardly homomallous or secund, plane leaf margins (which however can be narrowly recurved on one side in *D. humilis*), and a weakly differentiated costa, especially in *D. rufescens*. A unique character of *D. rufescens* among other ex-*Dicranella* species is the high basal membrane (up to 10 rows), while the membrane of *D. humilis* does not extend four rows; basal membranes extending three rows are nevertheless rare in all other species except *D. varia*. Both species markedly differ in their capsule shape (characteristically straight and symmetric in *D. rufescens*, while inclined, slightly curved, and asymmetric in *D. humilis*). It was therefore important to confirm that *D. humilis* is indeed closely related to *D. rufescens*, which we accomplished. All Asian specimens referred to *D. humilis* on the available morphological grounds appeared in the *D. rufescens* clade. Thus, although our study confirmed the species status of *D. humilis*, further morphological study of additional Asian specimens is needed to clarify the differentiation of *D. humilis* and *D. rufescens*. The isolated position of the clade precludes any other taxonomic solution except for establishing a new genus for the two species of the lineage, with the familial placement being somewhat ambiguous. The very weak clustering with the *Dicranella subulata + crispa* based in *trnK* data (Appendix A) might favor creating a family harboring both these lineages; however, the total evidence from all studied regions (Figure 1) does not support this solution and favors the creation of a separate family for this monogeneric lineage. This is supported by the salient morphological differences between the lineages (absent versus well-developed revoluble annulus, basal membrane 3–10 versus 1–3 rows, sparse leaves, never clasping and shouldered versus leaves dense and contiguous, at least perichaetial leaves clasping) in the absence of other nonhomoplasic common characters.

***Dicranella subulata* and *D. crispa*.** The morphological synapomorphies of this lineage were discussed at length by [10]. While the affinities with the preceding lineage have not been convincingly resolved, the same set of arguments can be used for segregating the two known representatives of this lineage to a separate genus and family (see Taxonomic treatment). The previous names adopted for *Dicranella crispa* and *D. subulata* include either names that are in use for other distinct genera, or the illegitimate genus names *Dicranodon* Béhéré and *Leptotrichum* Hampe ex Müll. Hal. Similarly, the possibility of raising *Dicranella* sect. *Pseudodicranella* Nyholm to the genus level is prevented by the name being invalid with respect to a missing Latin description and is illegitimate, as it includes the conserved type of *Dicranella*, *D. heteromalla*. Therefore, we propose to erect a new genus and family name for this group in the Taxonomy section.

**Dicranellaceae.** In agreement with [10], we concur with the proposal of reducing the delimitation of Dicranellaceae to only include members of the *Dicranella heteromalla* group, *Microcampylopus/Leptotrichella/Garckea/Campylopodium* polytomy, *Aongstroemia filiformis* s.lat., *Eccremidium,* and *Cladophascum*. With its remarkably distinct morphology [30], *Bryowijkia*, albeit robustly supported molecularly as a sister group to the above-specified assemblage, should remain separate at the family rank, as proposed by [31]. The question of the inclusion of *Trichodontium falcatum* (R. Br. bis) Fife remains open. The species was merged with *Kiaeria pumila* (Mitt.) Ochyra by [32], which was resolved as a member of *Arctoa* within Rhabdoweisiaceae [9]. However, the only available *Trichodontium* GenBank accessions AF435304/AF435353 from the specimen *Streimann* 51155 appeared in the clade with *Leptotrichella flaccidula* and *Campylopodium medium* in the analysis of [10], which suggests the possibility of an incorrect identification in one of the treatments, and the matter needs to be revisited in the future.

The maximally supported clade containing accessions of *Aongstroemia filiformis* s.lat. sister to the clade of other tropical Dicranellaceae also leaves us with no other option than to establish a new genus to accommodate it, as the species has never been included in genera other than *Aongstroemia*, *Dicranella*, *Dicranum,* and *Thysanomitrion,* which cannot be used for this purpose. This solution is put into effect later in the text. The geographically meaningful pattern of molecular variability, as supported by the analysis of an additional *Aongstroemia filiformis* specimen from Madagascar, seems to support the resurrection of *A. jamaicensis* from the synonymy of *A. filiformis,* but this task requires additional sampling and morphological study.

The well-supported tropical Dicranellaceae clade, consisting of analyzed accessions of *Aongstroemia filiformis* s.lat., *Microcampylopus*, *Leptotrichella*, *Garckea,* and *Campylopodium* also contains a single analyzed specimen of *Dicranella polii*. Its closest affinities were revealed to be with the previously analyzed *Garckea* species, with which it forms a clade moderately supported from ML and not supported from BI (BS 77–81, PP 0.86–0.88), nested within the well-supported clade containing the tropical Dicranellaceae, except for *Aongstroemia filiformis* s.lat. While the combination of *D. polii* under *Garckea* would make sense from a nomenclatural point of view, as the latter appears to be the oldest available generic name in this clade (*Leptotrichella* incl. the younger *Microdus*, *Microcampylopus*), the sporophytic characters currently used for delimitation between *Garckea*, *Leptotrichella* (generally considered synonymous to *Dicranella* [20,33,34], and *Microcampylopus* do not match the revealed phylogenetic affinities, and hence we prefer to postpone this taxonomic decision, pending a deeper sampling in this lineage. This brings, however, another piece of evidence that the tropical species referred previously to *Dicranella* s.lat., *Leptotrichella,* and *Microdus* might belong to this lineage, or to the lineage containing *Rhamphidium* species (see below for a discussion of *Dicranella vaginata*).

The affinities within *Dicranella* s.str. support the description of a new species, as realized below, and the continued recognition of *D. curvipes* from *D. heteromalla* at the specific rank, as suggested by [35], with the molecular support presented in a more limited dataset by [10]. Our additional data support the recognition of *Dicranella curvipes* as a separate entity, while further documenting the molecular variability within the lineage. Our review of gametophytic morphology revealed that, in some of *Dicranella curvipes* specimens, the leaves tend to be homomallous vs. mostly falcate secund in *D. heteromalla*; the costa is narrower (less than 1/5 of the leaf base width), well-delimited from leaf lamina (a unique trait in *Dicranella* s.str.), and unistratose throughout (vs. wider costa weakly delimited from leaf lamina, which is partly to nearly entirely bistratose distally in *D. heteromalla*); the cells in the basal leaf portion are narrowly rectangular and moderately thick-walled vs. short-rectangular to subquadrate, thin-walled in *D. heteromalla*). In addition, most specimens of *D. curvipes* have leaves with rather distinct shoulders. It needs to be acknowledged, though, that some specimens of *D. curvipes* (such as *Kučera* 21379, 21778) from the Russian Far East have other combinations of these characters and could not be identified without mature sporophytes. Moreover, there appears to be an internal differentiation of the clade consisting of plants currently assigned to *D. heteromalla* s.str., with the specimens from the Russian Far East (RF47, 49, FDt35, *Kučera* 21639) and one from the eastern United States (*Goffinet* 8162) showing several distinct molecular synapomorphies at the level of one-base substitutions. Although this clade is only weakly supported on the tree, with respect to the unequal sequenced regions in the studied accessions and ambiguous reads at several points, the lineage is probably molecularly distinct. The most diverged lineage in molecular terms is, however, the one harboring two accessions of plants collected on Iturup Island and originally identified as *Campylopus pyriformis*. Despite the few morphological traits differentiating these plants from *D. heteromalla*, the plants are described below as a morphologically semicryptic species; nevertheless, they are distinct with respect to their rate of molecular differentiation.

**Aongstroemiaceae.** The common characters of Aongstroemiaceae and features which differentiate the *Dicranella varia* group as the most alien element in Aongstroemiaceae were discussed in detail by [10]. Both morphology and molecular support for the clade containing *D. varia*, *D. varia* var. *obtusifolia*, *D. howei*, and *D. pacifica* require the generic rank to be used for this clade. In theory, the name *Anisothecium* could be applied to it, as *Anisothecium varium* is one of the six species cited in the protologue. However, we believe that this would be the least appropriate option for the typification of the genus, as [26] proposed this name in his “Musci Austro-Americani …” for a group of predominantly South American species, which mostly share vaginate or semivaginate leaves with distinctly widened leaf bases, while *Dicranella varia*, which was included based on a single specimen from Cuba, forms a distinctly discordant element in his circumscription of the genus, as was also emphasized in his key to the species. Therefore, we believe that *Anisothecium* is much more appropriately typified with one of the predominantly South American species with expanded leaf bases, as done below, and we propose a new genus name for the clade of *Dicranella varia* and closely related species.

Previous analysis [10] suggested that *Dicranella varia* was paraphyletic with respect to a specimen (RF42) from northern Siberia. Extended sampling of Asian material that was supposed to represent *D. varia* resulted in both plants being molecularly identical or closely related to specimens from Europe and plants identical to the previously studied RF42. The provenance of the latter specimens mostly included northern Siberia, while the lineage containing European plants included specimens collected throughout boreal Asia. Morphological examination of the north Siberian plants and comparison to *D. varia* s.str., as represented by both European specimens (the Central European lectotype from Leipzig, Germany, was reviewed by [36] and a specimen from the southern Siberia and the southern part of Russian Far East), confirmed the morphological differences between the two groups. It appeared that plants similar to the analyzed north Siberian ones had already been described. Lindberg and Arnell [37], who proceeded extensive bryophyte collections from the Russian Arctic, described *Anisothecium rubrum* var. *obtusiusculum* based on the plants from the lower course of the Yenisey River. Their description matches our plants well. They also mentioned that a similar taxon, *Dicranella varia* var. *obtusifolia* Berggren occurs in Svalbard and indeed provided a description that seems to match morphologically both the variety later described by [37] and the plants that we collected in northern Siberia. The type material held in MO (MO-407808, accession 2226886) shows a good match with the north Asian plants analyzed by us. Hence, we raise the variety earliest described by Berggren to the species rank in the newly established genus, as effected below in the Taxonomy section. The previously unsampled NW American endemic *Dicranella pacifica* W.R. Schofield, which shares with *D. varia* multiple characters including recurved leaf margins, inclined asymmetric capsules, and absent annulus [20,38], was confirmed as another member of this lineage. While the specific status of *D. howei,* which morphologically sometimes seems indistinct from *D. varia* [20], now appears unequivocal with respect to the resolved identity of *D. varia* var. *obtusifolia*, the elaboration of morphological differences remains the task for a future dedicated study with more numerous molecularly barcoded specimens.

The core Aongstroemiaceae clade contains two basal lineages, which were not sampled by [10]. The basalmost lineage is represented by a single collection of a plant from Sakhalin Island with a unique combination of the otherwise typical Aongstroemiaceae characters, including shouldered leaves; elongate laminal cells; a single costal stereid band and lack of guide cells; reddish setae; and short, dark, curved, smooth, or slightly furrowed exannulate capsules. On the contrary, the upper leaf lamina is bistratose to tristratose, a character that only occurs in some representatives of the family. This plant is therefore described below as a new monospecific genus. The second previously unsampled lineage is the likewise monotypic *Neodicranella*, whose affinities had not been well resolved in its protologue [23]. We were not able to convincingly assess its affinities, even now, due to the conflict in the resolved topologies between the LTRN and K datasets. Although the “dicranelloid” rather than “dichodontioid” habit would favor the affinities as assessed through analysis of the *trn*K data, we refrain, however, from merging *Neodicranella* with *Aongstroemia* at this point.

The crown clade of core Aongstroemiaceae contains the species recognized to date, aside from the above-mentioned conflicting position of *Neodicranella*, in at least four genera: *Aongstroemia* (type species *A. longipes*), *Hygrodicranum* (type species *H. falklandicum* Cardot, not analyzed), *Polymerodon* (monotypic), and *Dicranella* species with expanded, mostly vaginate leaf bases, which earlier were often assigned to *Anisothecium* (the analyzed *Dicranella campylophylla* is among the six *Anisothecium* species eligible for the type of the genus). The high molecular support for this clade, as well as the suite of morphological characters common in the analyzed taxa of this clade, strongly support the idea of recognizing this clade as one genus, for which the oldest available name is *Aongstroemia* (see below in the Taxonomy section). Within the genus, we were able to additionally analyze four specimens of *Dicranella schreberiana* s.str. from previously unsampled parts of Europe (Czech Republic, Greece, European Russia), which confirmed the distinctness from *D. schreberiana* var. *robusta*. Moreover, we morphologically revised the type specimen of *Cynodontium canadense* Mitt., which proved to be identical to *D. schreberiana* var. *robusta*. Having priority at species rank, *Cynodontium canadense* is combined under *Aongstroemia* later in the Taxonomy section, leaving however open the question of the specimen RF40, with the morphology rather suggesting *D. schreberiana* s.str., which was however found to be isolated from the clades representing both recognized varieties and did not form a monophylum with either of them. Additional analyzed specimens of *D. campylophylla* (including the specimens labelled as *D. cardotii* (R. Br. bis) Dixon, considered synonymous by [39] and subsequent authors) and *D. hookeri* brought more certainty to the taxonomic evaluation of these taxa. They were found to be closely related, yet probably distinct species, which however may be at times difficult to separate morphologically, as the accession *D. campylophylla* TJH13, which was unavailable for our study, was resolved in the polytomy formed by the accessions of *D. hookeri*, together with an accession labelled as *Polymerodon andinus*, which was downloaded from GenBank. The pattern is further complicated by the accession RF65, identified also as *D. hookeri*, which however is molecularly clearly distinct from all other accessions identified as this species, as well as from *D. campylophylla*. The type of *Hygrodicranum*, *H. falklandicum* Cardot remains unsampled, and hence the generic status of *Hygrodicranum* remains to be assessed. Based on the two analyzed accessions of *H. bolivianum* and one of *H. herrerae,* it appears rather safe to infer that both species are very closely related to *D. campylophylla*, yet possibly specifically distinct, although the protologue and illustrations of *Hygrodicranum herrerae* in [40] do not provide information that would distinguish this species from the descriptions and illustrations of *D. campylophylla* available in the literature [39]. The sequenced specimens IPG20 (*H. herrerae*), as well as TJH04 and TJH13 (*D. campylophylla* and *D. campylophylla/hookeri*) from Chile, are nearly identical, with bistratose leaf lamina and dorsally mamillose cells characteristic for both taxa. The specimen of *Hygrodicranum bolivianum* (*Buck* 39497), as well as the additionally studied Chilean specimen (*Larraín* 43529), matches the species description [41], which resembles some closely related *Dicranella* (*Aongstroemia*) species. Consequently, we propose to combine both *H. bolivianum* and *H. herrerae* in *Aongstroemia* and expect that *H. falklandicum* might be resolved in this clade as well, but until the species is analyzed, we prefer not to put this taxonomic change into effect. The same applies to the assessment of *Polymerodon andinus* (*rps4* and *nad5* sequences obtained from specimen *M. Lewis* 87608, 7/87 (DUKE) and *Eucamptodontopsis pilifera* (Mitt.) Broth. (*nad5* sequence obtained from specimen S.R. Hill 27912, 2/97 (DUKE)). If the sequences indeed correspond to these taxa, then they should be considered conspecific with *D. hookeri,* but in the absence of type studies and a more representative selection of analyzed material, such a proposal is premature, as further corroborated by the affinities of GenBank sequences of *Eucamptodontopsis brittoniae* (E.B. Bartram) B.H. Allen (AF435285, AF435328), which appear to be closely related to *Holomitrium* species based on BLAST searches.

The caution with the taxonomic evaluation of this group of taxa can be illustrated by the example of *Dicranella vaginata.* This species was considered to be closely related to the group of South Hemispheric *Dicranella* species recognized as *Anisothecium* by [26] in the protologue of the genus. However, three Chilean accessions analyzed by us were found to be resolved in the maximally supported clade containing two *Rhamphidium* species and *Symblepharis krausei* (Lorentz) Ochyra & Matteri. Indeed, all species have the vaginate leaf base, elongate basal leaf cells with porose longitudinal cell walls, subquadrate upper leaf cells, mostly unistratose upper leaf lamina, and rather short erect or inclined, nearly symmetric capsule with markedly spiculose, deeply divided peristome teeth, different from the typical dicranoid, i.e., not spiculose, and less divided peristome shared by *Aongstroemia* species. Recent morphological studies found *Dicranella vaginata* very similar to *Aongstroemia gayana* [42,43], which further indicates the necessity of a modern polyphasic reassessment of the lineage containing the type of *Rhamphidium*.

Consequently, we propose to typify the genus *Anisothecium* with *A. campylophyllum*, which in our opinion best preserves Mitten’s original idea to include in it mostly South American representatives of the then recognized broad genus *Aongstroemia* with broadened, mostly clasping leaf bases and dicranoid affinities. Neither *A. varium* nor *A. vaginatum* (see above) seem to qualify better for this purpose. The identity of *A. jamesonii* Mitt. is currently ambiguous, as it was considered synonymous either to *A. vaginatum* [44] or to *Dicranella hookeri* [45], which were not found to be closely related by us, and hence a new investigation of the type is necessary in light of this finding. Similarly, we have no molecular data for the remaining South American species of the original *Anisothecium*, *A. convolutum* (Hampe) Mitt. and *A. planinervium* (Taylor) Mitt.

The polyphyly of *Aongstroemia*, *Dicranella*, and *Ditrichum* demonstrated by the analyses in [8,10] and this study is yet another example of homoplasy of morphological characters that were considered taxonomically relevant in earlier classifications. The superficially similar small pioneer mosses that are adapted to opportunistic life strategies sometimes occupy a remarkably isolated phylogenetic position among the basal lineages of Dicranidae. They share with most other protohaplolepideous lineages the broad, typically circumholarctic ranges, usually associated with humid climates, suggesting that the early diversification of Dicranidae was associated with cool to mild conditions, and therefore might be underestimated in temperate and subarctic, and by analogy possibly also in subantarctic, areas. The early radiation might not have been followed by excessive diversification according to our current knowledge, but the pioneer strategy of their representatives might have allowed them to survive until the present.

In contrast, the later diverging lineages of (mostly) opportunistic pioneer mosses (Aongstroemiaceae s.str., Dicranellaceae s.str., Ditrichaceae s.str. and some groups of Pottiaceae) are remarkably more diverse, in terms of species numbers, morphologically, and ecologically, often occupying xeric environments (such as several groups of Pottiaceae) and displaying multiple transitions to annual life strategies. Although only a limited number of *Aongstroemia* and especially *Dicranella* species from outside the Holarctic have been studied, the preliminary *rps4*-based phylogenetic analysis of Brazilian *Dicranella* s.lat. species indicates that the studied Neotropical *Dicranella* species all belong to the lineage of Dicranellaceae [46].

Our results demonstrate the unexpectedly underestimated diversity of northern temperate and subarctic pioneer mosses with dicranelloid habit and the resulting limitations of the currently used floras, especially in North Asia. In addition, northern Asia is an area of higher molecular diversity of *Dicranella* s.l. species, while European accessions are typically uniform in sequences, which might indicate the role of northeastern Asia as a source of diversity in these lineages worldwide.

The stunning extent of convergence in the available morphological traits within the studied genera underlines the need for obtaining molecular data for the representatives of the as yet unevaluated taxa and also the revision of types for existing names. Given the number of poorly known taxa (>600 names in *Dicranella* and >260 in *Aongstroemia*, [22]), such a project would require the efforts of the whole bryological community.

## 4. Taxonomy

**Rhizogemmaceae** Bonfim Santos, Siebel & Fedosov, **fam. nov.**–Type: *Rhizogemma* Bonfim Santos, Siebel & Fedosov

**Diagnosis:** This family differs from the other families of haplolepideous mosses in possessing the following combination of characters: plants small to medium-sized; stems with central strand; leaves with widened leaf bases abruptly narrowed to short subulate leaf tips; leaf margins recurved; costae with central stereid band, dorsal and ventral epidermis, without guide cells; laminal cells elongate, smooth; sexual condition dioicous; setae yellow; capsules asymmetric, furrowed; peristome dicranoid, bright-red; annulus revoluble.

The family is currently considered monogeneric.

***Rhizogemma*** Bonfim Santos, Siebel & Fedosov, **gen. nov.**–Type: *Rhizogemma staphylina* (H. Whitehouse) Bonfim Santos, Siebel & Fedosov (Figure 2 and Figure 3).

**Diagnosis:** The single species segregated into the newly established genus differs from other dicranelloid mosses in possessing non-vaginate leaf bases and rather shortly subulate leaf acumina, recurved leaf margins, costae with single central stereid band, leaf lamina unistratose or bistratose along upper margins, yellow setae, yellow-purplish to brownish, asymmetric, furrowed capsules, revoluble annulus, and rhizoidal gemmae irregular in shape, composed of bulging cells.

**Etymology:** The name (composed of the Greek ῥίζα, root, and Latin *gemma,* gem) refers to the characteristic rhizoidal tubers (commonly also referred to as gemmae) of the only currently known species of the genus.

**Description:** Plants bright green, lacking red pigmentation. Stems about 5 mm, forming rather dense tufts, with a central strand. Leaves up to 1 mm long, lanceolate, erect-spreading to spreading, not secund; margins plane or recurved only at base or nearly throughout, smooth or denticulate distally, partly bistratose distally; costae percurrent to short excurrent, in transverse section with differentiated dorsal and ventral epidermis and single stereid band; leaf lamina unistratose, cells rectangular to elongate-rectangular, bulging in transverse sections, smooth. Rhizoidal tubers constantly present, in young stage red, turning dark brown, irregularly shaped with protruding cells, 3–4 cells long and 1–3 cells wide. Perichaetial leaves differentiated, larger than lower leaves, from broadly sheathing base rather abruptly narrowed into squarrose or flexuose apex. Setae yellowish, straight; capsules incurved, longitudinally furrowed, without or weak strumae, yellow to purplish along ribs and around the mouth. Exothecial cells irregular in shape to rectangular, with evenly incrassate walls. Annulus well differentiated, composed of one row of large hyaline thick-walled cells, revoluble. Operculum long rostrate. Peristome bright red, teeth unequally split to half of their length, longitudinally striolate proximally, papillose distally. Calyptrae cucullate.

The genus is currently considered monotypic.

***Rhizogemma staphylina*** (H. Whitehouse) Bonfim Santos, Siebel & Fedosov, **comb. nov.** ≡ *Dicranella staphylina* H. Whitehouse in Trans. Brit. Bryol. Soc. 5: 757. f. 2-3a. 1969.–Type: United Kingdom, E. Norfolk (v.-c. 27), Pockthorpe, near Lyng, arable field, Sept. 1968, *H.L.K. Whitehouse s.n.* (Holotype: CGE) ≡ *Anisothecium staphylinum* (H. Whitehouse) Sipman, Rubers & Riemann in Lindbergia 1: 217. 1972.

**Ruficaulaceae** Bonfim Santos & Fedosov, **fam. nov.**–Type: *Ruficaulis* Bonfim Santos & Fedosov

**Diagnosis:** This family differs from the other families of haplolepideous mosses in possessing the following combination of characters: small plant size; mature stems reddish-brown; costae with only dorsal stereid band; leaf lamina unistratose or with bistratose margin, composed of elongate to linear cells; sexual condition dioicous; red setae; peristome dicranoid, with rather weak ornamentation, well-developed basal membrane and weakly developed annulus.

The family is currently considered monogeneric.

***Ruficaulis*** Bonfim Santos & Fedosov **gen. nov.**–Type: *Ruficaulis rufescens* (With.) Bonfim Santos & Fedosov

**Etymology:** The generic name originates from the Latin *caulis*, stem, and the prefix *rufi-* (from Latin rufus, red), referring to the characteristic reddish-brown coloration of the stem.

**Diagnosis:** species combined in the newly established genus differ from other dicranelloid mosses in possessing reddish-brown mature stems; leaves from narrow-triangular base gradually narrowed to a subulate acumina; costae with single stereid band; leaf lamina unistratose or with bistratose margins, composed of elongate to linear cells; red setae; peristome with mostly high basal membrane and weakly developed annulus.

**Description:** Plants very small, in loose reddish-brownish tufts. Stems with central strand. Well-developed parts of stems reddish-brown and rhizoids with vinaceous pigmentation. Leaves up to 2 mm long, fuscous, weakly secund; margins plane throughout, denticulate at apex; costa rather weak, percurrent, sharply delimited from leaf lamina, with compact stereid band, ventral and dorsal epidermis, or with weakly developed dorsal band, composed of substereids and guide cells covered by ventral epidermis. Tubers consisting of one row of (1–)2–3(–6) much enlarged cells, pale reddish to wine-red. Perichaetial leaves similar to upper leaves. Setae reddish. Capsules erect to inclined, symmetric or curved, smooth or slightly furrowed. Exothecial cells short rectangular, in longitudinal rows. Annulus weakly differentiated, not revoluble. Peristome dicranoid, with high basal membrane.

Accepted species:

***Ruficaulis rufescens*** (With.) Bonfim Santos & Fedosov, **comb. nov.** ≡ *Bryum rufescens* With. in Syst. Arr. Brit. Pl. (ed. 4) 3: 801. 1801–Type: “ad ripas rivulorum lutosas, in Scotia” ≡ *Dicranum rufescens* (With.) Turner in Muscol. Hibern. Spic. 66. 1804 ≡ *Dicranum varium* var. *rufescens* (With.) Röhl. in Deutschl. Fl. (ed. 2), Kryptog. Gew. 3: 71. 1813 ≡ *Dicranodon varium* var. *rufescens* (With.) Béhéré in Muscol. Rothom. 29. 1826 ≡ *Dicranum crispum* var. *rufescens* (With.) Hampe in Flora 20: 283. 1837 ≡ *Aongstroemia rufescens* (With.) Müll. Hal. in Syn. Musc. Frond. 1: 436. 1848 ≡ *Dicranella rufescens* (With.) Schimp. in Coroll. Bryol. Eur. 13: 1856 ≡ *Anisothecium rufescens* (With.) Lindb. in Musci Scand. 26. 1879.

***Ruficaulis humilis*** (R. Ruthe) Jan Kučera & Fedosov, **comb. nov.** ≡ *Dicranella humilis* R. Ruthe, Hedwigia 12: 147. 1873.–Type: [Germany] In locis paucis prope Bärwalde Neomarchicae. ≡ *Anisothecium humile* (R. Ruthe) Lindb., Meddeland. Soc. Fauna Fl. Fenn. 14: 74. 1887 ≡ *Aongstroemia humilis* (R. Ruthe) Müll. Hal., Gen. Musc. Frond. 323. 1900. ≡ *Dicranella rufescens* subsp. *humilis* (R. Ruthe) Kindb., Eur. N. Amer. Bryin. 2: 209. 1897.

**Dicranellopsidaceae** Bonfim Santos, Siebel & Fedosov, **fam. nov.**–Type: *Dicranellopsis* Bonfim Santos, Siebel & Fedosov

**Diagnosis:** This family differs from the other families of haplolepideous mosses in possessing the following combination of characters: plants small to medium-sized; stems with central strand; leaves with widened to vaginate leaf bases, abruptly narrowed to subulate leaf acumina; leaf margins plane; costae with two stereid bands and guide cells; leaf lamina bistratose distally; laminal cells elongate, smooth; sexual condition dioicous; red setae; ribbed capsules; Dicranoid peristome and revoluble annulus.

The family is currently considered monogeneric.

***Dicranellopsis*** Bonfim Santos, Siebel & Fedosov, **gen. nov.**–Type: *Dicranellopsis subulata* (Hedw.) Bonfim Santos, Siebel & Fedosov

**Etymology:** The generic name originates from *Dicranella*, the genus to which the species of *Dicranellopsis* had been assigned previously, and the suffix -opsis (from Greek ὄψις, meaning aspect, appearance, sight), referring to the morphological similarity between the genera.

**Diagnosis:** Species combined in the newly established genus differ from other dicranelloid mosses in possessing a combination of widened to vaginate leaf bases and subulate leaf tips, plane leaf margins, costae with two stereid bands and guide cells, bistratose upper leaf lamina, red setae, ribbed capsules and revoluble annulus.

**Description:** Plants yellowish green to dark green, lacking red pigmentation. Stems with central strand. Leaves with oblong bases abruptly tapering into a long, channeled, subulate acumina, upper stem leaves sheathing, erect to squarrose-flexuose, patent or secund, crispate or not when dry; margins entire or very slightly denticulate at leaf tip, plane, unistratose; costae percurrent to short excurrent, sharply delimited from leaf lamina, with dorsal and ventral epidermis, guide cells and dorsal and ventral or only dorsal stereid band; distal leaf lamina bistratose, median leaf cells linear. Rhizoidal tubers, when present, dark brown, irregularly shaped without protruding cells, curved. Perichaetial leaves resemble upper stem leaves. Capsules erect to slightly inclined, symmetric or distinctly curved, not strumose, strongly longitudinally ribbed, with more or less differentiated exothecial bands and quadrate to short rectangular, rather thin-walled cells between them. Annulus differentiated in 2–3 rows of widened cells, revoluble. Peristome dicranoid.

Accepted species:

***Dicranellopsis crispa*** (Hedw.) Bonfim Santos, Siebel & Fedosov, **comb. nov.** ≡ *Dicranum crispum* Hedw. in Sp. Musc. Frond. 132. 1801–Lectotype: Sweden, J.F. Ehrhart s.n. (G, barcode G00040017, [36]: Figure 1B–D; [47]: Figure 51) ≡ *Aongstroemia crispa* (Hedw.) Müll. Hal. in Syn. Musc. Frond. 1: 439. 1848 ≡ *Dicranella crispa* (Hedw.) Schimp. in Coroll. Bryol. Eur. 13. 1856 ≡ *Leptotrichum crispum* (Hedw.) Mitt. in J. Proc. Linn. Soc., Bot., Supplementary 1: 158. 1859 ≡ *Cynodontium crispum* (Hedw.) Mitt. in J. Proc. Linn. Soc., Bot. 8: 15. 1864 ≡ *Anisothecium crispum* (Hedw.) C.E.O. Jensen in Skand. Bladmossfl. 314. 1939, nom. illeg., non Lindb. in Utkast Eur. Bladmoss. 33. 1878.

***Dicranellopsis subulata*** (Hedw.) Bonfim Santos, Siebel & Fedosov, **comb. nov.** ≡ *Dicranum subulatum* Hedw. in Sp. Musc. Frond. 128. T. 34. f. 1–5. 1801–Lectotype: Sweden, Swartz s.n. (G, Hb. Hedwig-Schwägrichen, barcode G00040102, [36]: Figure 5A–C, [47]: Figure 61) ≡ *Dicranodon subulatum* (Hedw.) Béhéré in Muscol. Rothom. 29. 1826 ≡ *Dicranum heteromallum* var. *subulatum* (Hedw.) Wallr. in Fl. Crypt. Germ. 1: 160. 1831 *nom. illeg.* ≡ *Aongstroemia subulata* (Hedw.) Mül. Hal. in Syn. Musc. Frond. 1: 433. 1848 ≡ *Dicranella subulata* (Hedw.) Schimp. in Coroll. Bryol. Eur. 13: 1856 ≡ *Leptotrichum subulatum* (Hedw.) Mitt., in J. Proc. Linn. Soc., Bot., Supplementary 1: 9. 1859 ≡ *Cynodontium subulatum* (Hedw.) Mitt. in J. Proc. Linn. Soc., Bot. 8: 15. 1864.

**Dicranellaceae** Stech in Nova Hedwigia 86: 14. 2008–Type: *Dicranella* (Müll. Hal.) Schimp.

Accepted genera: *Campylopodium* (Müll. Hal.) Besch. (only *C. medium* studied), *Cladophascum* Dixon, *Dicranella* (Müll. Hal.) Schimp. *Eccremidium* Wilson (only *E. floridanum* studied), *Garckea* Müll. Hal., *Leptotrichella* (Müll. Hal.) Lindb. (only *L. flaccidula* studied), *Microcampylopus* (Müll. Hal.) Fleisch., *Bryopalisotia* Bonfim Santos & Fedosov.

Tentatively included genus (pending molecular confirmation): *Bryotestua* Thér. & P. de la Varde. As for *Trichodontium* (Dixon) Fife, see the Discussion.

**Description:** Plants small to medium-sized, growing in turfs. Acrocarpous or cladocarpous. Central strand present. Leaves appressed or erect-spreading, often flexuose or falcate-secund, narrowly lanceolate, often subulate. Lamina cells rectangular, smooth, not porose. Alar cells not differentiated. Costa single, strong, (sub-)percurrent to (long) excurrent, with guide cells, dorsal and ventral stereid bands and differentiated ventral and dorsal epidermis. Dioicous or autoicous. Seta elongate, erect, sinuose or arcuate, or short, erect or curved. Capsule erect to horizontal or pendulous, symmetric or gibbous, occasionally strumose, smooth or plicate, ovoid to short-cylindric with operculum conic to long-rostrate, or globose with operculum dome-shaped to hemispheric with a blunt apiculus. Stomata present or absent. Peristome dicranoid or absent. Spores usually papillose, sometimes warty. Calyptra cucullate or mitrate.

The following synopsis only includes the genera where taxonomic novelties are proposed.

***Dicranella*** (Müll. Hal.) Schimp. in Coroll. Bryol. Eur. 13. 1856–Type: *Dicranella heteromalla* (Hedw.) Schimp.

Accepted species: *Dicranella cerviculata* (Hedw.) Schimp., *D. curvipes* (Lindb.) Ignatov, *D. heteromalla* (Hedw.) Schimp., *D. thermalis* Fedosov & Ignatova (see below).

Excluded species: *Dicranella campylophylla* (Taylor) A. Jaeger, *D. crispa* (Hedw.) Schimp., *D. grevilleana* (Brid.) Schimp., *D. hookeri* (Müll. Hal.) Cardot, *D. howei* Renauld & Cardot, *D. humilis* Ruthe, *D. pacifica* W.B. Schofield, *D. riparia* (H. Lindb.) Mårtensson & Nyholm, *D. rufescens* (With.) Schimp., *D. schreberiana* (Hedw.) Hilf. ex H.A. Crum & L.E. Anderson, *D. staphylina* H. Whitehouse, *D. subulata* (Hedw.) Schimp., *D. varia* (Hedw.) Schimp.

Species with uncertain placement: all other accepted species (cf. [22]), pending morpho-molecular studies, including *Dicranella polii* Renauld & Cardot and *D. vaginata* (Hook.) Cardot, for which our molecular phylogenetic data suggest placement outside *Dicranella* as recognized here, but additional sampling is needed to assign the generic affinities, as discussed above.

***Dicranella thermalis*** Fedosov, Ignatova & Jan Kučera, **sp. nov.** (Figure 4).

**Diagnosis:** The new species resembles *D*. *heteromalla* in the rather robust plant size, non-shouldered leaves, wide costae occupying up to ½ of the leaf width and weakly delimited from leaf lamina, with thin-walled cells with large lumen forming ventral surface of costa in basal leaf portion, but differs from it in having homomalous rather than falcate secund leaves and weakly serrulate to nearly entire upper leaf margins.

**Type:** Russia, Sakhalin Province, Iturup Island. South-West slope of Baranskogo Volcano, Goryachaya River. *Fedosov & Pisarenko* 19 September 2015, Mosses of the Russian Far East Exsiccatae No. 78 (as *Campylopus pyriformis*). (Holotype: MW: MW9090383, Isotypes: MHA, NSK, VGBI, MO, NY).

**Etymology:** The species name refers to the typical habitat of the species at the type locality.

**Description:** Plants medium-sized, stems up to 3 cm, single, with well-developed central strand, evenly foliate, tomentose in lower part. Leaves more or less appressed when dry, spreading when wet, gently falcate-secund, 2.5–3.2 × 0.25–0.35 mm, widest at base, from lanceolate base gradually tapering into canaliculate subulate acumen; margins plane, unistratose, weakly and bluntly toothed throughout or only in upper half, near apex with double teeth: costae strong, occupying 1/3–1/2 of leaf base, rather indistinctly delimited from the leaf lamina, with one row of guide cells, two stereid bands, and differentiated dorsal and ventral epidermis; sometimes ventral epidermis immediately covering guide cells or guide cells forming surface of costa ventrally; leaf lamina partly or completely bistratose distally, upper leaf cells 24–38 × 5–6 µm, elongate-rectangular, smooth, moderately thick-walled; basal leaf cells of the same length and 8–11 µm wide. Sexual condition and sporophytes unknown.

**Differentiation:** We did not find more characteristics to differentiate this molecularly distinct species from *Dicranella heteromalla* than those specified in the diagnosis. *D. thermalis* resembles *Campylopus pyriformis* (Schultz) Brid. in having wide costae, undifferentiated alar regions, and thin-walled cells with wide lumen on a ventral surface in the basal portion of leaf. In contrast to most *Campylopus* species, *D. thermalis* possesses two stereid bands.

**Ecology and distribution**: The species is known from numerous collections (held mostly in MW) on the slope of Baranskogo volcano in Iturup Island (45.07° N, 147.98° E), where it grows along the hot stream banks under *Sasa* understory at altitudinal range of 220–280 m a.s.l. Similar non-sporulating *Dicranella* plants were frequently encountered in thermal habitats of Kamchatka Peninsula and northern part of Kunashir Island, but they were mostly not collected and therefore their identity remains uncertain.

Paratypes (the same locality, date and collectors as in the holotype): Accession numbers MW9073555-MW9073558, MW9007288-MW9007292, MW9073559.

***Bryopalisotia*** Bonfim Santos & Fedosov, **gen. nov.**–Type: *Bryopalisotia filiformis* (P. Beauv.) Bonfim Santos & Fedosov

**Etymology:** The name was chosen as a tribute to A.M.F.J. Palisot, Baron de Beauvois (1752–1820), a French naturalist and author of Prodrome des cinquième et sixième familles de l’Æthéogamie, les mousses, les lycopodes [48], in which the type species of the genus was described as *Dicranum filiforme* P. Beauv.

**Diagnosis:** This genus differs from *A. longipes* and several other species of *Aongstroemia* in its traditional circumscription in its robust habit, leaves with sheathing leaf base, abruptly narrowed into a long, subulate leaf apex, and elongate to linear, extremely thick-walled basal leaf cells. From the genus *Aongstroemia* in its newly proposed circumscription, *Bryopalisotia* differs in having cylindric rather than ovoid or shortly ellipsoid capsules. Elongate to linear, extremely thick-walled basal leaf cells differ *Bryopalisotia* from *A. guayana*.

The genus is presently considered monospecific, although the below-stated synonymy should be revisited (see Discussion).

***Bryopalisotia filiformis*** (P. Beauv.) Bonfim Santos & Fedosov, **comb. nov.** ≡ *Dicranum filiforme* P. Beauv. in Prodr. Aethéogam. 53. 1805–Type: Isle de Bourbon [=Réunion], *Bory s.n.* ≡ *Thysanomitrion filiforme* (P. Beauv.) Arn. In Mém. Soc. Linn. Paris 5: 263. 1827 ≡ *Aongstroemia filiformis* (P. Beauv.) Wijk & Margad. in Taxon 9: 50. 1960 = *Aongstroemia jamaicensis* Müll. Hal., Bull. Herb. Boissier 5: 554, 1897 *fide* [49].

**Aongstroemiaceae** De Not. in Atti Reale Univ. Genova 1: 30. 1869–Type: *Aongstroemia* Bruch & Schimp.

Accepted genera: *Aongstroemia* Bruch & Schimp., *Calcidicranella* Bonfim Santos, Fedosov & Jan Kučera, *Dichodontium* Schimp., *Diobelonella* Ochyra, *Neodicranella* Porley & Fedosov, *Protoaongstroemia* Fedosov, Ignatova & Jan Kučera.

Tentatively included genus (pending molecular confirmation): *Aongstroemiopsis* M. Fleisch. Genera tentatively moved in synonymy (see below): *Hygrodicranum* Cardot, *Polymerodon* Herzog.

Plants minute to medium-sized, in loose to dense turfs. Stems julaceous or not, central strand present. Stem leaves with a broad sheathing base tapering into a blunt apex or abruptly narrowed to a short or long acumen. Margins entire, crenulate or weakly denticulate to dentate. Leaf lamina 1–2(–3) stratose; laminal cells variable in shape, usually smooth but mamillose or papillose in some species. Alar cells not differentiated. Costa subpercurrent to mostly short to long excurrent, weak to strong. Asexual reproduction via gemmae (on filamentous branches at the leaf axils) or rhizoidal tubers. Dioicous. Seta elongate, straight or flexuose. Capsule variable in shape, ovoid to curved and sometimes slightly strumose, smooth or furrowed when dry, operculate, with peristome teeth vertically pitted-striolate at base. Annulus not or poorly differentiated. Operculum conic or rostrate. Calyptra cucullate.

The following synopsis only includes genera where taxonomic novelties are proposed.

***Calcidicranella*** Bonfim Santos, Fedosov & Jan Kučera, **gen. nov.**–Type: *Calcidicranella varia* (Hedw.) Bonfim Santos, Fedosov & Jan Kučera.

**Etymology:** The generic name originates from the generic name *Dicranella*, where this species has been placed for a long time, and the prefix *calci-* referring to the ecological preference for calcareous substrates in the species included in the genus.

**Diagnosis:** Species combined in the newly established genus differ from other dicranelloid mosses in possessing non-vaginate leaf bases, partly to nearly entirely recurved leaf margins, smooth laminal cells, costae with well-differentiated stereids in one or two bands, red setae, dark reddish-brown, asymmetric, inclined capsules, and non-revoluble annulus, and by its ecological preference for base-rich mineral soil.

**Description:** Central strand present. Leaves lanceolate, gradually narrowed to blunt, acute or acuminate apex, without sheathing base, margins recurved on one or both sides; costa weakly or rather sharply delimited from leaf lamina, typically with guide cells and two or rarely only dorsal stereid band, differentiated dorsal and, in several species, also ventral epidermis; leaf lamina unistratose or with bistratose patches to entirely bistratose distally; leaf cells rectangular. Rhizoid tubers occasionally present, irregular in shape, with protruding cells, 100–140(–250) × 60–95 μm. Dioicous. Perichaetial leaves similar to lower leaves. Setae red. Capsules inclined, asymmetric, ovoid, gibbous, smooth or furrowed when dry, dark red when mature. Exothecial cells irregular in shape or rectangular, with thickened longitudinal walls. Annulus weakly differentiated, not revoluble. Peristome dicranoid.

***Calcidicranella howei*** (Renauld & Cardot) Bonfim Santos, Fedosov & Jan Kučera, **comb. nov.** ≡ *Dicranella howei* Renauld & Cardot in Rev. Bryol. 20: 30. 1893.–Type: [United States of America], Cal. [=California], *M.A. Howe.*

***Calcidicranella varia*** (Hedw.) Bonfim Santos, Fedosov & Jan Kučera, **comb. nov.** ≡ *Dicranum varium* Hedw. in Sp. Musc. Frond.: 133. 1801–Lectotype: [Germany, Leipzig], [Hedwig?] s.n. (G, Hb. Hedwig-Schwägrichen, barcode G00040364, [36]: Figure 5D–F; [47]: Figure 77) ≡ *Dicranodon varium* (Hedw.) Béhéré in Muscol. Rothom. 29. 1826 ≡ *Aongstroemia varia* (Hedw.) Müll. Hal. in Syn. Musc. Frond. 1: 435. 1848 ≡ *Dicranella varia* (Hedw.) Schimp. Coroll. Bryol. Eur. 13: 1856 ≡ *Anisothecium varium* (Hedw.) Mitt. in J. Linn. Soc., Bot. 12: 40. 1869.

***Calcidicranella pacifica*** (W.B. Schofield) Jan Kučera & Fedosov, **comb. nov.** ≡ *Dicranella pacifica* W.B. Schofield, Bryologist 73: 703, 1970.–Holotype: Canada. British Columbia: Vancouver, Spanish Banks, 49°16′ N, 123°14′ W, seepy silt cliffs and cliff base, *Schofield* 40,422 (UBC).

***Calcidicranella obtusifolia*** (Berggren) Fedosov, Ignatova & Jan Kučera, **comb. et stat. nov.** ≡ *Dicranella varia* var. *obtusifolia* Berggren, Kongl. Svenska Vetensk. Acad. Handl., n.s. 13(7): 36. 1875–Type: Musci Spetsbergens. Exsicc. No. 9. Figure 5 ≡ *Anisothecium varium* var. *obtusifolium* (Berggr.) Podp., Consp. Musc. Eur. 118. 1954.

=*Anisothecium rubrum* var. *obtusiusculum* Lindb. & Arnell, Kongl. Svenska Vetensk. Acad. Handl., 23(10): 85. 1890, **syn. nov**.–Type: ‘Fl. Jen., T. subarct., Polovinka fr.’ [Flora Jeniseensis, subarctic Taimyr, vicinity of Polovinka River] ≡ *Dicranella varia* var. *obtusiuscula* (Lindb. & Arnell) Paris, Index Bryol. 336. 1896 ≡ *Anisothecium varium* var. *obtusiusculum* (Lindb. & Arnell) Podp., Consp. Musc. Eur. 118. 1954.

**Description**: Plants small, gregarious, light green or yellowish. Stems simple, ca. 0.1–0.2 cm, with strong round central strand and weak sclerodermis, evenly foliated. Leaves appressed, straight or slightly curved when dry, spreading when moist, 1.0–1.8(–2.2) mm, with wide, ovate bases and more or less distinct shoulders, above shoulders gradually narrowed towards blunt acumen, concave, lower leaves not widened, triangular; margins plane at base, narrowly recurved at shoulders and just above and below them or almost to the leaf tip, unistratose proximally and partly bistratose distally, uneven above, rarely throughout the margin; costae ending just below apices, rarely percurrent, rather strong, occupying ca. 1/7–1/5 of leaf base, distinctly delimited from leaf lamina, in transverse section with 2–4(–5) large ventral guide cells, differentiated dorsal epidermis and single weak stereid band; leaf lamina unistratose with occasional bistratose strands distally; upper leaf cells short rectangular to subquadrate, 12–20 × 7–12 µm, smooth, not bulging, proximally longer and wider, 44–90 × 10–17 µm, elongate-rectangular, 2–3 rows of cells along margins narrower, ca. 4–6 µm wide. Dioicous. Perichaetial leaves of the same length, but with wider and longer base, more abruptly narrowed to lanceolate or short subulate acumen. Setae 3–5 mm red to brownish. Capsules ca. 1 mm, asymmetric, curved, ovate, with short neck, strumose, brownish-red, distinctly furrowed, red rimmed distally, exothecial cells irregular in shape, thick-walled with equally thickened walls, longer and narrower along furrows, with few stomata proximally. Annulus not differentiated. Operculum conic. Peristome teeth red to brownish, 450–500 µm long, unequally split for nearly half of their length, longitudinally striolate proximally, papillose distally. Spores 14–17 µm, smooth, yellowish-brown, mature in summer. Rhizoidal tubers not seen.

**Differentiation**: *C. obtusifolia* resembles *C. varia* or *C. howei* in habit but differs in smaller plants with stems up to 5 mm, while stems of *C. varia* often extend to 1 cm. Leaf margins in *C. obtusifolia* are plane below shoulders, while *C. varia* has leaf margins recurved from the basal leaf portion and *C. howei* has leaf margins recurved mostly in the lower leaf part only, often only on one side. Leaf tips in *C. obtusifolia* are typically blunt, with costae ending a few cells below tips to being percurrent, while in *C. varia/howei* leaf tips are sharp and costae excurrent. Capsules of *C. obtusifolia* are strumose and distinctly longitudinally furrowed, while in *C. varia/howei* capsules are not strumose, smooth or rarely indistinctly furrowed. Exothecial cells in *C. obtusifolia* approach *C. howei*, they are irregular in shape, with equally thickened walls, while in *C. varia* longitudinal walls of exothecial cells typically are thicker than transverse ones. Although in many formal characters *C. obtusifolia* resembles North American *C. pacifica*, the latter species is much larger; moreover, with its contorted to crisped leaves and smooth capsules it is quite distinct from *C. obtusifolia*.

**Distribution and ecology:** A predominantly Arctic species, described from Svalbard and also known from a single locality in Nenets Autonomous District (European Russia), suite of localities along Yenisey River, in Taimyr Peninsula, Anabar Plateau and from a single locality in Yakutia. According to the protologue of *Anisothecium rubrum* var. *obtusiusculum* [37], it is also one of the most frequent mosses along the Yenisey River banks, although it rarely occurs in sufficient amounts, while in Svalbard it is either rare or not recognized from *C. varia*. It grows on bare loamy soil and silty sediments including saline ones on eroded slopes along rivers and in massives of baidzarakhs (thermokarst mounds), most often with *Hennediella heimii* var. *arctica*, *Funaria* spp., *Tortula leucostoma*, *T.* cf. *cernua, Bryoerythrophyllum* spp., *Aloina brevirostris, Stegonia latifolia, Pohlia atropurpurea, Bryum* spp., and many other pioneer mosses. At the same time, according to our field experience, it differs from other *Dicranella* s.l. species widespread in Siberian Arctic in occupied habitats, since these usually settle on acidic sandy sediments, typically with gemmiferous species of *Pohlia, Pogonatum* and *Psilopilum* species.

***Protoaongstroemia sachalinensis*** Fedosov, Ignatova & Jan Kučera, **gen. et spec. nov.**–Type: Russian Far East, Sakhalin Island, Tym’ River valley, 50.89518° N, 142.65693° E, in silty alluvium, 4 September 2009, *O.Yu. Pisarenko* op03352, MHA (Holotype), MW, NSK 2,003,352 (Isotypes) (Figure 6).

**Etymology:** The generic name originates from *Aongstroemia* (a genus of dicranoid mosses) and the prefix *proto-* (from Greek πρῶτος, first), which reflects the basalmost position of the genus within the core Aongstroemiaceae clade. The specific epithet reflects the provenance of the original collection, the Sakhalin Island.

**Diagnosis:** Differs from other Holarctic *Dicranella* s.l. species by the combination of distinctly shouldered leaves, distally regularly 2–3-stratose lamina, costa with a single stereid band and undifferentiated guide cells, elongate rectangular laminal cells and irregularly furrowed, curved capsules.

**Description:** Plants small, gregarious, light green or yellowish, mixed with other pioneer mosses. Stems simple, ca. 0.1–0.2 cm, with central strand and weak sclerodermis, evenly foliated. Leaves appressed, straight or slightly curved when dry, spreading when moist, gradually increasing in size distally, 1.5–1.9 × 0.4–0.53 mm, with wide, ovate base, widest at ca. 1/10–1/5 of leaf length with distinct shoulders, abruptly narrowed into gradually tapering blunt acumen, concave; margins plane, with few blunt distant teeth at shoulders and upper part of acumen to nearly entire, plane, partly bistratose proximally; costa weak, weakly delimited from leaf lamina, percurrent, in transverse section with ventral and dorsal epidermis and single band of substereids between them, without guide cells proximally, weakly differentiated distally; leaf lamina unistratose with bistratose strands proximally, 2–3 stratose distally; leaf cells elongate-rectangular, 37–62 × 6–13 µm, smooth, bulging on both sides, proximally somewhat longer, 45–75 µm long. Dioicous, male plants not seen. Perichaetial leaves with wider base, abruptly narrowed to short subulate acumen. Setae reddish, 5–7 mm, spirally twisted when dry and moving around after wetting. Capsules 1.2–1.5 mm long, asymmetric, curved, ovate, with short neck, weakly furrowed, not strumose, reddish-brown, red-rimmed distally; exothecial cells rectangular, moderately thick-walled with evenly incrassate transverse and longitudinal walls, longer and narrower along furrows, with few stomata in proximal part. Annulus not differentiated. Operculum conic or with short blunt oblique beak. Peristome teeth bright red, ca. 300 µm, unequally split for nearly half of their length, longitudinally striolate proximally, papillose distally. Spores 13–17 µm smooth, yellowish-brown, mature in autumn. Rhizoidal tubers not seen.

**Differentiation:** With its shouldered and then gradually narrowed leaves, elongate rectangular laminal cells and short curved capsules, *P. sachalinensis* habitually resembles a small *Diobelonella,* especially Asian populations with narrower leaves. However, it differs not only in its size but also in having bistratose leaf lamina. The same trait and plain margins differentiate *P. sachalinensis* from the somewhat similar *Calcidicranella varia*. Among species with partially bistratose lamina, *P. sachalinensis* differs from *Dicranellopsis subulata* in its non-subulate distal leaf portion, lack of guide cells and undifferentiated annulus; it differs from *Calcidicranella pacifica* in having shouldered leaves, narrower and longer leaf cells and lack of guide cells; and from *C. howei* in shouldered leaves and narrower costa.

**Distribution and ecology:** This newly described species is known from a single specimen, which was collected on silty alluvium sediments of Tym’ River in the middle part of Sakhalin Island. This pioneer moss grew together with *Ruficaulis* cf. *rufescens*, *Ceratodon purpureus* and male plants of *Pohlia* cf. *lescuriana*.

***Aongstroemia*** Bruch & Schimp. in Bryol. Eur. 1: 171 (fasc. 33-36. Mon. 1). 1846, *nom. & orth. cons. ‘Angstroemia’.* Type: *Aongstroemia longipes* (Sommerf.) Bruch & Schimp. [50]

=*Anisothecium* Mitt., J. Linn. Soc., Bot. 12: 39, 1869, **syn. nov.**–Type: *Anisothecium campylophyllum* (Taylor) Mitt., J. Linn. Soc., Bot. 12: 40. 1869, **designated here.**

=Dicranella p.pte., Hygrodicranum Cardot p.pte.

? = *Eucamptodontopsis* Broth. p.pte.

? = Polymerodon Herzog. 

Note: Since *Dicranella* now has a conserved type, *D. heteromalla* (Hedw.) Schimp. [51], the name *Anisothecium* should no longer be considered illegitimate. As argued above, the best candidate to typify the name is *Anisothecium campylophyllum* with respect to the good match with the general intent of the author and known phylogenetic affinities of this species.

Diagnostic characters: Stem leaves with a broad sheathing base tapering into a blunt apex (in less developed *A. longipes* plants) or abruptly narrowed to short or long pointed, spreading to squarrose leaf apex. Lamina cells rectangular, smooth or sometimes mamillose or papillose, sometimes (irregularly) bistratose. Tubers, if present, spherical without protruding cells. Capsules erect to inclined, symmetric to asymmetric, oval/obloid to curved and sometimes slightly strumose, on a straight, erect, red to brownish seta. Annulus not or poorly differentiated.

For a list of accepted species see below.

Excluded species: *Aongstroemia filiformis* (P. Beauv.) Wijk & Margad. (see above under *Bryopalisotia*).

Species with uncertain placement: all other accepted species (cf. [22]), pending morpho-molecular studies, and also *Aongstroemia orientalis* Mitt., for which molecular phylogenetic data [10] suggest placement in Ditrichaceae, but additional sampling is required to assess its affinities within this family.

***Aongstroemia boliviana*** (Herzog) Bonfim Santos & Fedosov, **comb. nov.** ≡ *Hygrodicranum bolivianum* Herzog in Biblioth. Bot. 87: 15. pl. 1: f. 1. 1916–Type: [Bolivia] Glazialtümpel am Cerro Incachacca, ca. 4600 m, No. 2599; an Steinen im Bach, oberes Llavetal, ca. 4200 m, No. 4832; in einem Quellbach des Pajonaltales, ca. 4000 m, No. 3264; in einem Quellbach der Cerros de Malaga, ca. 4000 m, No. 4359.

***Aongstroemia campylophylla*** (Taylor) Müll.Hal. in Syn. Musc. Frond. 2: 608. 1851. ≡ *Dicranum campylophyllum* Taylor in London J. Bot. 7: 281. 1848–Lectotype (designated in [39,45]): 8 Aug. 1847 W. Jameson 133 (BM000879353, Isolectotypes BM0006722168, BM000879354) ≡ *Anisothecium campylophyllum* (Taylor) Mitt. in J. Linn. Soc., Bot. 12: 40. 1869 ≡ *Dicranella campylophylla* (Taylor) A. Jaeger in Ber. Thätigk. St. Gallischen Naturwiss. Ges. 1870-71: 382 (Gen. Sp. Musc. 1: 86). 1872.

=*Dicranum cardotii* R.Br. bis in Trans. & Proc. New Zealand Inst. 35: 329. 36 f. 9. 1903, *fide* [50].–Type: [New Zealand], “on damp banks, tributary of the River Hapuka, near Kaikoura” Robert Brown s.n. (Holotype: BM-Dixon [52]). ≡ *Dicranella cardotii* (R.Br. bis) Dixon in New Zealand Inst. Bull. 3(3): 77. 1923 ≡ *Anisothecium cardotii* (R. Br. bis) Ochyra in Moss Fl. King George Island Antarctica 114. 1998.

=*Cheilothela vaginata* H. Rob. fide [45] = *Dicranella convoluta* (Hampe) A. Jaeger fide [45] = *Symblepharis tenuis* R.S. Williams fide [45].

***Aongstroemia canadensis*** (Mitt.) Siebel & Fedosov, **comb. nov.** ≡ *Cynodontium canadense* Mitt., Proc. Linn. Soc., Bot. 8: 17. 1864–Type: [Canada] British N. America (probably from the Rocky Mountains) T. Drummond, no 101 in part, (probable holotype NY325565) ≡ *Dicranella canadensis* (Mitt.) Austin in Bot. Gaz. 2: 96. 1877 ≡ *Dichodontium canadense* (Mitt.) Lesq. & James in Man. Mosses N. America 62. 1884.

=*Dicranella schreberi* var. *robusta* Schimp. ex Braithw. in J. Bot. 9: 289. 1871, **syn. nov.**–Type: [United Kingdom, England], at various places in Cheshire, at Milnthorpe (Barnes) and near Melnrose (c. fr., Jerdon) [specimen Rabenhorst, Bryotheca Europaea No. 74 mentioned as bearing well-developed sporophytes ≡ *Dicranella schreberiana* var. *robusta* (Schimp. ex Braithw.) H.A. Crum & L.E. Anderson].

*=Anisothecium schreberianum* var. *elatum* (Schimp.) Wijk & Margad. in Taxon 7: 288. 1958, *fide* [53].

The identity of *Dicranella schreberiana* var. *robusta*, treated under this name by [10], with the type of *Cynodontium canadense* was suggested by H. Siebel (pers. comm.), who prepared a detailed account on this taxon.

***Aongstroemia grevilleana*** (Brid.) Müll. Hal. in Syn. Musc. Frond. 1: 439. 1848 ≡ *Dicranum schreberi* var. *grevilleanum* Brid. in Bryol. Univ. 1: 450. 1826–Type: [UK] In humidis argillaceis Scotiae. *Greville, Hooker, Arnott s.n.* ≡ *Dicranum grevilleanum* (Brid.) Bruch & Schimp. in Bryol. Eur. 1: 123. 54 (fasc. 37-40. Mon. 19. 7.). 1847 ≡ *Dicranella grevilleana* (Brid.) Schimp. in Coroll. Bryol. Eur. 13. 1856 ≡ *Anisothecium grevilleanum* (Brid.) Arnell & C.E.O. Jensen in Bih. Kongl. Svenska Vetensk.-Akad. Handl. 21 Afd. 3(10): 49. 1896 ≡ *Dicranella schreberi* var. *grevilleana* (Brid.) Mönk. in Laubm. Eur. 179. 1927.

***Aongstroemia herrerae*** (R.S. Williams) Bonfim Santos & Fedosov, **comb. nov.** ≡ *Hygrodicranum herrerae* R.S. Williams in Bryologist 29: 37. pl. 3: f. 1–9. 1926 (‘*herrerai*’, cf. ICN Art. 60.8.a)–Type: “Growing about waterfalls, Río Tapfi, province of Cuzco, Peru, at 3600 m.” *F.L. Herrera* No. 792, Sept. 1925, same locality, *F.L. Herrera* No. 798a”.

***Aongstroemia hookeri*** Müll.Hal., Syn. Musc. Frond. 2: 607. 1851.–Type: Insula Eremitae ad Cap. Horn: *J.D. Hooker*. ≡ *Anisothecium hookeri* (Müll. Hal.) Broth., Nat. Pflanzenfam. (ed. 2) 10: 178. 1924. ≡ *Dicranella hookeri* (Müll. Hal.) Cardot, Bull. Herb. Boissier, sér. 2, 6: 4. 1906.

=*Anisothecium perpusillum* Dusén fide [54] = *Dicranella subclathrata* Lorentz fide [55] = Meesia patagonica Dusén fide [55].

? = *Polymerodon andinus* Herzog, Beih. Bot. Centralbl., 26(2): 48. pl. 1. 1909.–Type: Bolivia: An feuchten Felsen neben dem Weg im Valle de Llave (bei Cochabamba), ca. 3600 m, mit *Wollnya stellata* Herzog: Januar, 08.

? = *Eucamptodontopsis pilifera* (Mitt.) Broth., Nat. Pflanzenfam. (ed. 2) 10: 202. 1924. ≡ *Eucamptodon pilifer* ‘*piliferus*’ Mitt., J. Linn. Soc., Bot., 12: 69. 1869.–Type: Trinidad, Margarita, Palma Real, *Crüger*.

***Aongstroemia longipes*** (Sommerf.) Bruch & Schimp., Bryol. Eur. 1: 173 (fasc. 33–36 Monogr. 3). 1846. ≡ *Weissia longipes* Sommerf., Suppl. Fl. Lapp. 52, pl. 1, f. 1–10. 1826.–Type: “In terra argillosa humida ad rivulos montanos in provincia Saltdalen Norvegiae (*Sommerfelt* s.n.); in Canada superiore (*Drummond* s.n.)”.

***Aongstroemia schreberiana*** (Hedw.) Bonfim Santos & Fedosov, **comb. nov.** ≡ *Dicranum schreberianum* Hedw. in Sp. Musc. Frond. 144, pl. 33, f. 6-10. 1801–Lectotype: [Germany, Saxony, Leipzig], sin. coll. s.n. (G, Hb. Hedwig-Schwägrichen, barcode G00040018, [36]: Figure 3F–H; [47]: Figure 73) ≡ *Anisothecium schreberianum* (Hedw.) Dixon in Rev. Bryol. Lichénol. 6: 104. 1934 ≡ *Dicranella schreberiana* (Hedw.) Hilf. ex H.A. Crum & L.E. Anderson in Mosses E. N. Amer. 1: 169. 1981 = *Bryum crispum* Schreber, *nom. inval.*, Spic. Fl. Lips. 79 (no. 1038). 1771 = *Dicranum schreberi* Sw. *nom. illeg.*, Monthly Rev. 34: 538. 1801.

## 5. Materials and Methods

### 5.1. Taxon Sampling

The matrix of molecular data was largely based on that used for the backbone phylogeny of Dicranidae, with a focus on *Dicranella* and *Aongstroemia* [10]. With respect to the absence of dicranelloid taxa in some lineages of haplolepidous mosses, we reduced the matrix by leaving out or reducing the number of accessions in lineages where these taxa were absent, in order to decrease the complexity of the alignment. The outgroups were thus reduced to include only *Pseudoditrichum*, *Flexitrichum*, *Scouleria*, *Drummondia*, *Hymenoloma,* and *Bryoxiphium*, and we further substantially reduced the representation of Leucobryaceae, Rhabdoweisiaceae, Dicranaceae, and related families (leaving out completely *Mittenia*, *Pleurophascum*, *Serpotortella*, Hypodontiaceae, Octoblepharaceae, and Calymperaceae), and also Ditrichaceae including *Aongstroemia orientalis* and *A. julacea*, which will be treated in a dedicated future article. On the other hand, we added accessions of *Dicranella staphylina*, *D. humilis*, *D. varia* incl. its neglected var. *obtusifolia*, *D. pacifica*, *D. grevilleana*, *D. schreberiana* incl. its var. *robusta*, *D. campylophylla*, *D. hookeri*, *D. heteromalla*, *D. curvipes*, *D. polii*, *D. vaginata*, *Aongstroemia longipes*, *A. filiformis*, *Neodicranella hamulosa,* and unassigned dicranelloid plants from Pacific Russia, which were found to be related to *Dicranella* s.str. and to *Diobelonella*/*Dichodontium*/*Neodicranella* grade. Newly generated *trnK–psbA* data were added for at least one representative of each major lineage left. Laboratory protocols for isolation of DNA, amplification and sequencing followed the protocols described in [8,10,56,57]. Genbank accession numbers of the included specimens and vouchers of specimens studied de novo are compiled in Appendix B.

### 5.2. Phylogenetic Analyses

Sequences were aligned using MAFFT v. 7 [58] with the E-INS-i strategy and otherwise default settings, and the resulting alignment was improved manually at obviously misaligned sites. The concatenated dataset (available in http://purl.org/phylo/treebase/phylows/study/TB2:S30163, accessed on 28 January 2023) was tentatively partitioned according to the sequenced regions (*trnF–trnL*, abbreviated L hereafter, *trnL–rps4* (T), *rps4–trnS* (R), *trnK–psbA* (K), *nad5* (N) with respect to their significantly differing coverage, rather than according to coding and non-coding regions. The best-fit partitioning scheme and models of nucleotide evolution were searched for in PartitionFinder2 [59]. The results of the greedy algorithm used suggested partitioning according to all of the initially suggested partitions, with the HKY + I+G model for the *trnF–trnL* partition and GTR + I+G for the remaining ones. Indel data were scored for individual partitions using the simple indel coding (SIC) approach [60] in SeqState 1.4.1 [61] and added to the dataset in three variants: (1) indels scored only for L, R, and N partitions; (2) indels scored for L, T, R, and N partitions; and (3) indels scored for all partitions. Based on the results of [10], we did not separately analyze the L, R, and N data with respect to the reported absence of conflicts in topology and relatively low resolution of trees obtained from single-gene analyses, but we explored the influence of previously unused regions, i.e., (a) data from spacers flanking *trnT* between *trnL* and *rps4* (T) and (b) *trnK–psbA* (K) data, which were successively added to the working pilot analyses. Given the amount of phylogenetic signal, the K data were also analyzed separately from the 52 accessions for which these data were available.

Phylogenetic reconstructions were performed using Bayesian inference (BI) and maximum likelihood (ML). BI was run in MrBayes v.3.2.7 [62] in two parallel runs, each consisting of eight Markov chains run for 2,000,000 generations as default, and with further generations added if the convergence between runs did not reach 0.01, with the default number of swaps and a sampling frequency of one tree for each 100 generations. The chain temperature was initially set at 0.1 and lowered as necessary according to the acceptance rates. The models were sampled throughout the GTR model space and gamma-distributed rate variation across sites, and a proportion of invariable sites, as suggested by the PartitionFinder. PSRF values, were checked as being close to 1.000. ESS values were checked using Tracer v.1.7.2 [63] as being higher than 200. Consensus trees were calculated after omitting the burn-in of the first 25% of trees. The best-scoring maximum likelihood (ML) trees were searched using the new rapid hill-climbing algorithm in RAxML 8.2.12 [64] under the GTR model with gamma model of rate heterogeneity in 50 independent runs, each starting from a different random tree. The extended majority-rule consensus tree criterion was used to stop the bootstrapping used for the assessment of the node robustness. Analyses were performed using the grid computational services provided by the MetaCentrum Virtual Organization (see Acknowledgement). Trees were visualized using TreeGraph2 [65].

### 5.3. Morphological Studies

In addition to standard microscopic observations during the revision of herbarium specimens, preparation of taxon descriptions, and illustrations, images of peristomes were obtained by scanning electron microscopy (SEM) with a JSM-6380 (JEOL) at the User Facilities Center of M.V. Lomonosov Moscow State University. Peristomes mounted on stubs were coated with gold without any additional preparation, and light microscope illustrations were made under a stereomicroscope Olympus SZX-7 with a digital camera Infinity 8, with Z-stacking in Helicon Software [66].

## Figures and Tables

**Figure 1 plants-12-01360-f001:**
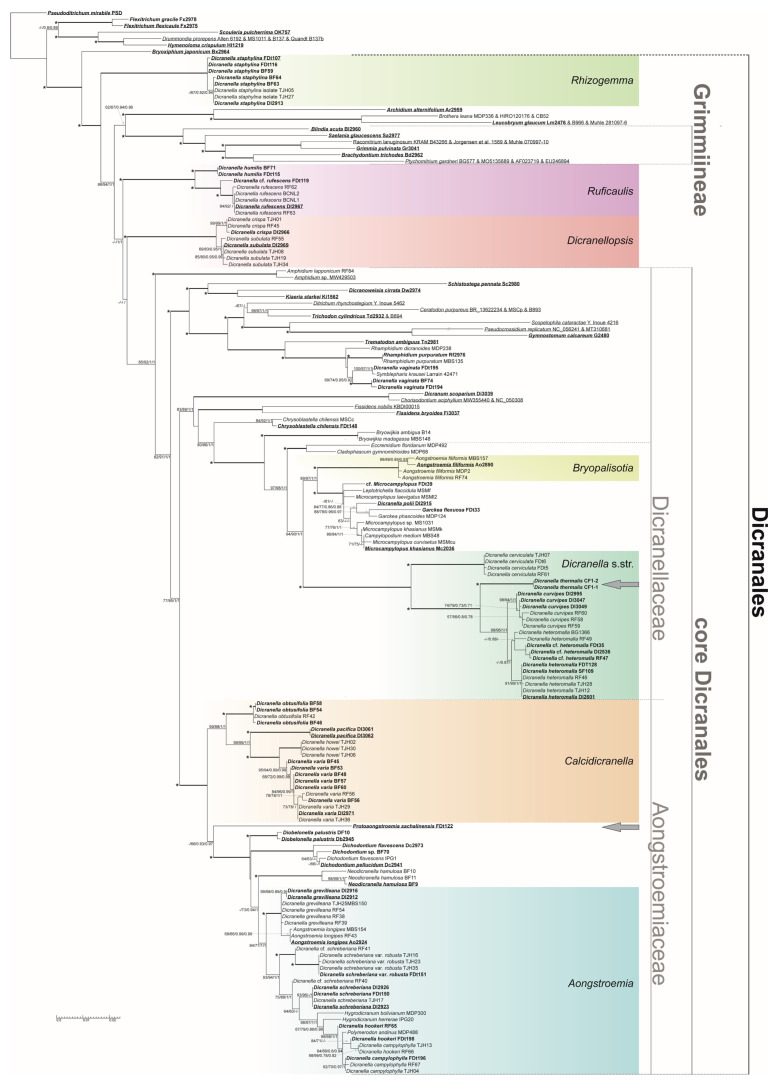
Maximum likelihood tree, inferred from the concatenated data matrix from the chloroplast *trnF–trnS* and *trnK–psbA* and mitochondrial *nad5* intron 2 sequence data (LTRKN dataset) of selected species of Dicranidae, focused on the genera *Dicranella* and *Aongstroemia*, rooted with *Pseudoditrichum mirabile*. Bootstrap support values higher than 60 inferred from ML analyses, without and with indel coding, and posterior probabilities higher than 0.7 inferred from BI, without and with indel coding, are shown above the branches; hyphens in place of support values denote lower support of the node, while a blank space indicates that the node is absent from the topology inferred from the particular analysis; maximally supported nodes are indicated by solid lines and asterisks. Newly studied terminals, as well as terminals for which at least one marker was obtained de novo, are printed in bold, and terminals for which the *trnK–psbA* sequence is available are underlined. For details, see Appendix B.

**Figure 2 plants-12-01360-f002:**
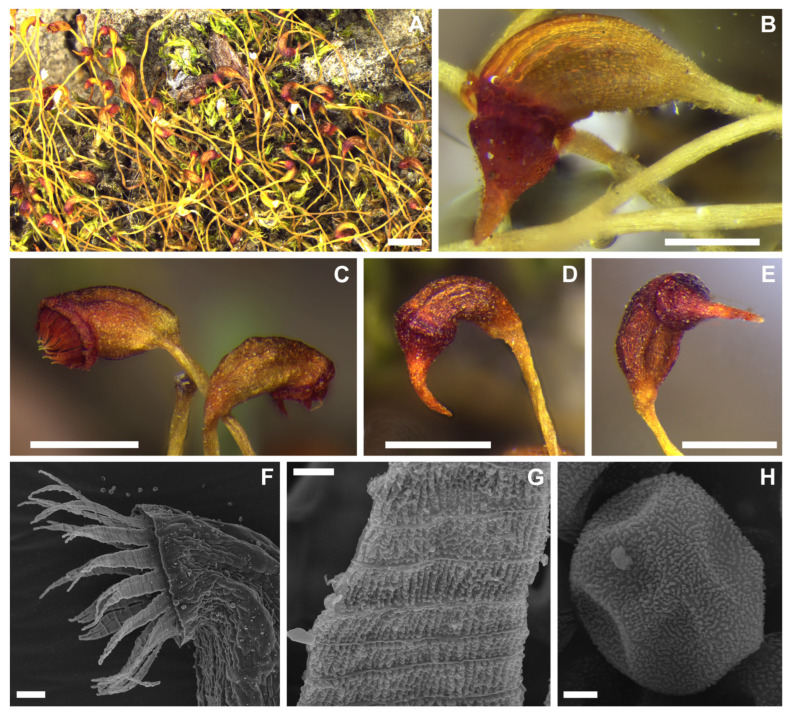
Sporophytes and peristome of *Dicranella staphylina* (from North Siberia, MW9117945, FDt107). (**A**) fragment of specimen with sporophytes, (**B**) wet opening capsule with dehiscent annulus, (**C**–**E**) dry capsules, (**F**) SEM image of peristome, (**G**) outer surface of the lower part of segment, (**H**) spore. Scale bars: 1 mm for (**A**); 0.5 mm for (**B**–**E**); 100 μm for (**F**); 10 μm for (**G**); 2 μm for (**H**).

**Figure 3 plants-12-01360-f003:**
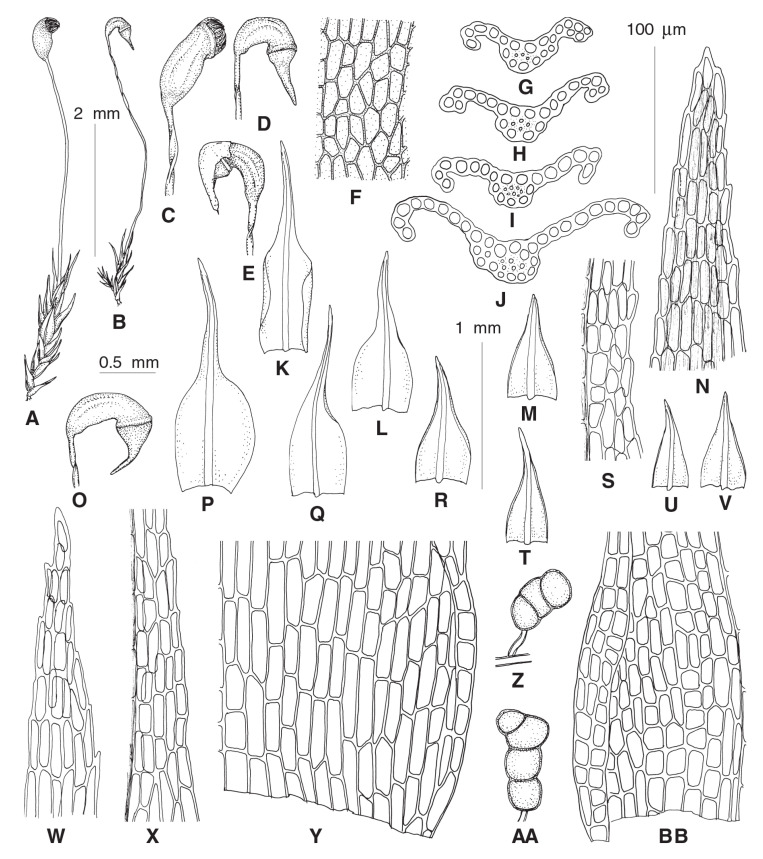
Line drawings of gametophyte and sporophyte of Siberian plants of *Rhizogemma staphylina* (from Russia: Krasnoyarsk Territory, Putorana Plateau, MW9117945, isolate FDt107): (**A**) fertile plant, wet, (**B**) fertile plant, dry, (**C**–**E**,**O**) capsules, dry, (**F**) exothecial cells, (**G**–**J**) leaf transverse sections, (**K**,**P**) perichaetial leaves, (**L**,**M**,**Q**,**R**,**T**,**U**,**V**) stem leaves, (**N**,**W**) upper-leaf cells, (**S**,**X**) mid-leaf cells, (**Y**,**BB**) basal leaf cells, (**Z**,**AA**) gemmae. Scale bars: 2 mm for (**A**,**B**); 0.5 mm for (**C**–**E**,**O**); 1 mm for (**K**–**M**,**P**–**R**,**T**,**U**,**V**); 100 μm for (**F**–**J**,**N**,**S**,**W**–**BB**).

**Figure 4 plants-12-01360-f004:**
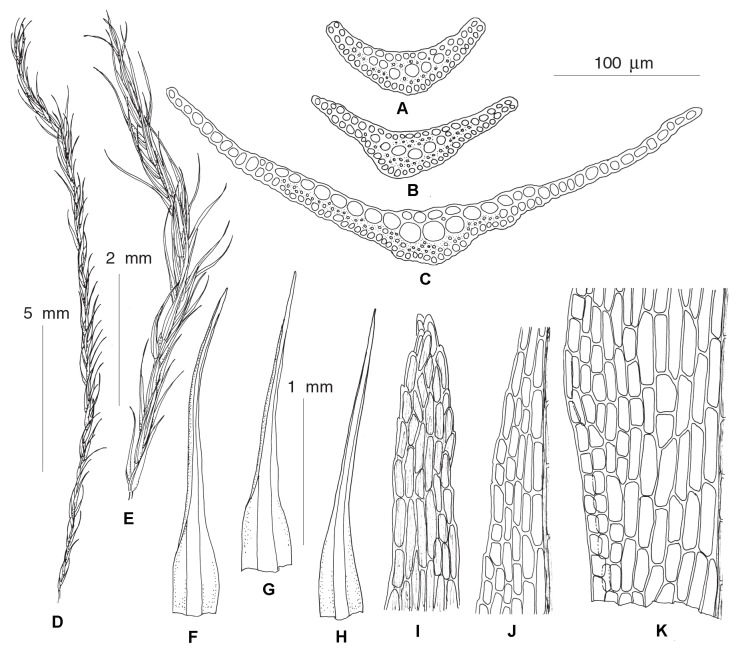
Line drawings of gametophyte of *Dicranella thermalis* (from: Holotype, isolate CF1-1): (**A**–**C**) leaf transverse sections, (**D**,**E**) view of sterile plants, (**F**–**H**) leaves, (**I**) upper-leaf cells, (**J**) mid-leaf cells, (**K**) basal leaf cells. Scale bars: 5 mm for (**D**); 2 mm for (**E**); 1 mm for (**F**–**H**); 100 μm for (**A**–**C**,**I**–**K**).

**Figure 5 plants-12-01360-f005:**
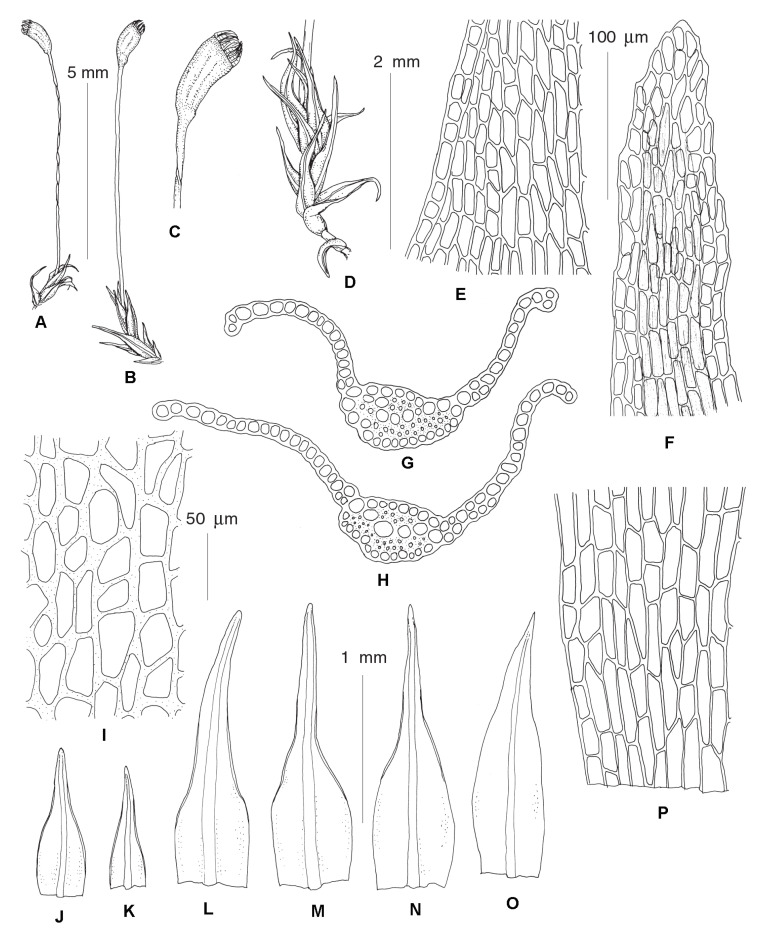
Line drawings of gametophyte and sporophyte of *Calcidicranella obtusifolia* (from: Russia, Krasnoyarsk Territory, Anabar Plateau, MW9031184, isolate RF42): (**A**,**D**) view of fertile plants, dry, (**B**) view of fertile plant, wet, (**C**) capsule, dry, (**E**) mid-leaf cells, (**F**) upper-leaf cells, (**G**,**H**) leaf transverse sections, (**I**) exothecial cells, (**J**–**N**) stem leaves, (**O**) perichaetial leaf, (**P**) basal leaf cells. Scale bars: 5 mm for (**A**,**B**); 2 mm for (**C**,**D**); 1 mm for (**J**–**O**); 100 μm for (**E**,**F**,**I**,**P**).

**Figure 6 plants-12-01360-f006:**
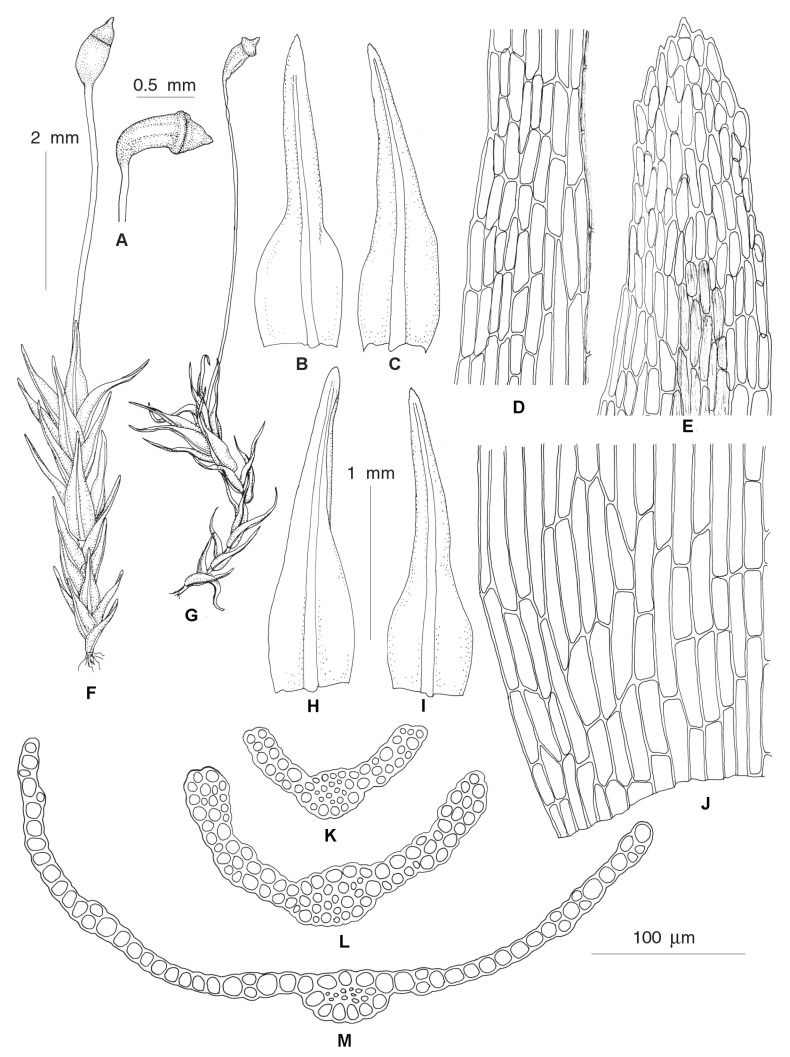
Line drawings of gametophyte and sporophyte of *Protoaongstroemia sachalinensis* (from: Holotype, isolate FDt122): (**A**) capsule, dry, (**B**) perichaetial leaf, (**C**,**H**,**I**) stem leaves, (**D**) mid-leaf cells, (**E**) upper-leaf cells, (**F**) view of fertile plant, wet, (**G**) fertile plant, dry, (**J**) basal leaf cells, (**K**–**M**) leaf transverse sections. Scale bars: 2 mm for (**F**,**G**); 1 mm for (**B**,**C**,**H**,**I**); 0.5 mm for (**A**); 100 μm for (**D**,**E**,**J**–**M**).

## Data Availability

All authors agree with MDPI Research Data Policies.

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
