# Peer review of "New Taxonomic Arrangement of Dicranella s.l. and Aongstroemia s.l. (Dicranidae, Bryophyta)"

_plants, 2023, doi:10.3390/plants12061360_

Round 1

Reviewer 1 Report

Dear Editors, 

The manuscript ‘New taxonomic arrangement of Dicranella s.l. and Aongstroemia s.l. (Dicranidae, Bryophyta)’ brings more information on the evolution of studied groups. I'm not so familiar with bryology, nevertheless, the phylogeny presented looks to be valid in my point of view. The manuscript is well written, in general, and may be published, however has some weakness, indicated below.

One important point, in my opinion, is the lack of information on biparental inheritance. Why did the authors not include ITS marker of rDNA, for instance, to the study? Now it is hard to say whether the phylogenies based on ptDNA+mtDNA and ncDNA congruent or not. It should be discussed somehow. Despite the fact that hybridization has been well described for angiosperms, apparently, it should matter for bryophytes. I think this point should be covered in more detail.

Thus, if the authors have no chance to present data based on ncDNA markers, it should be emphasized that all conclusions are based on only maternal inheritance.

The major weakness of the manuscript is the limitations of the information presented in M&M part.

First of all, I recommend dividing the M&M into different paragraphs, e.g. plant collection, DNA isolation, PCR and sequencing, phylogenetic analysis, microscopy etc. Right now, the information in each part is very limited to estimate the eligibility of using approaches. For instance, laboratory protocols for isolation of DNA, amplification and sequencing were essentially the same as described in [8, 10, 27, 28]. What does exactly mean 'were essentially the same': were the same or almost the same? If they were the same, why not list only one Ref., over vice it is not clear now which special combination of the methods the authors used from the cited studies in the present manuscript? I strongly recommended describing the molecular approaches even briefly with an appropriate citation for each approach used.

It seems that the used voucher information, sampling points, and Genbank accession numbers are absent. The authors wrote ‘Genbank accession numbers of the included 1076 specimens and vouchers of specimens studied de novo are compiled in Appendix 1’. However, supplementary materials do not contain any information mentioned. That is why it is hard to say the originality of the results of the present study.

I think the map of the collection sites would be also helpful.

A little more about supplementary materials. I’ve found only one file with only one tree without any figure caption, that is why it is hard to say what is especially described on that. Nevertheless, in the text there are several linkages to supplementary files (e.g., Figures S1 and S2, Appendix 1), but it looks that materials are skipped on their own. Thus, it is not possible to check the propriety of the description of some parts of the text at the moment. These points need to be confirmed and all figures must be presented in the manuscript. 

In M&M part the information about indels coding is also limited. It seems that that information is important because the authors applied a different strategies of analyzing these dataset (lines 1086-1095). However, the information of how many indels, their positions in alignment, those were coded is omitted. The best practice is to upload alignments in the public data repository, e.g. TreeBASE, especially if the indels information used for the phylogeny reconstructing.

Please, also clarify which substitutional model was used for indel datasets.

Because of quite informed discussion, I would strongly recommend presenting the conclusion as a separate section. Right now, the ‘Concluding remarks’ presents as a subsection of the discussion. It seems the conclusion can be prepared based on the aforementioned subsection but more concise, or prepare de novo.

The resolution of Figure 1 is too poor, the node support levels as well as some taxon names are not readable.

The correspondent author is not indicated and the section of the author contributions is omitted.

I think the English polishing would also improve the manuscript.

Author Response

The manuscript ‘New taxonomic arrangement of Dicranella s.l. and Aongstroemia s.l. (Dicranidae, Bryophyta)’ brings more information on the evolution of studied groups. I'm not so familiar with bryology, nevertheless, the phylogeny presented looks to be valid in my point of view. The manuscript is well written, in general, and may be published, however has some weakness, indicated below.

One important point, in my opinion, is the lack of information on biparental inheritance. Why did the authors not include ITS marker of rDNA, for instance, to the study? Now it is hard to say whether the phylogenies based on ptDNA+mtDNA and ncDNA congruent or not. It should be discussed somehow. Despite the fact that hybridization has been well described for angiosperms, apparently, it should matter for bryophytes. I think this point should be covered in more detail.

Thus, if the authors have no chance to present data based on ncDNA markers, it should be emphasized that all conclusions are based on only maternal inheritance.

---With respect to the polyphyly of Dicranella and Aongstroemia in its previous delimitation, the presented phylogeny spans a phylogenetically very diverse group, nearly the whole subclass of Dicranidae (haplolepidous mosses) with more than 4,000 estimated species. While it would be surely ideal to reconstruct the phylogeny using both nuclear and organellar data from a genomically representative set of markers, the budget for this study has not allowed for a wider genomic sampling and no well-performing nuclear markers have ever been developed to reconstruct phylogeny on the subclass-spanning level that we have studied (the available and broadly used hypervariable markers such as ITS produce very unreliable alignments, while LSU and SSU when used alone typically produce little resolved trees even on such a relatively high taxonomic rank). Fortunately, there is a high level of congruence between phylogeny based on nuclear and organellar markers at higher taxonomic levels, as shown, e.g., by Liu et al. 2019 (https://doi.org/10.1038/s41467-019-09454-w); hybridization typically does not affect relationships at the high levels of taxonomy where gene flow among taxa is not significant. We have nevertheless added a note explaining these facts to the last paragraph of Introduction.

The major weakness of the manuscript is the limitations of the information presented in M&M part.

First of all, I recommend dividing the M&M into different paragraphs, e.g. plant collection, DNA isolation, PCR and sequencing, phylogenetic analysis, microscopy etc.

--- We accepted this suggestion, subheadings added.

Right now, the information in each part is very limited to estimate the eligibility of using approaches. For instance, laboratory protocols for isolation of DNA, amplification and sequencing were essentially the same as described in [8, 10, 27, 28]. What does exactly mean 'were essentially the same': were the same or almost the same? If they were the same, why not list only one Ref., over vice it is not clear now which special combination of the methods the authors used from the cited studies in the present manuscript? I strongly recommended describing the molecular approaches even briefly with an appropriate citation for each approach used.

---Multiple references for obtaining of molecular markers were mentioned as we used a broader set of markers than in our previous studies and sometimes the amplification protocols are marker-specific and each of the papers describes techniques used to obtain the particular marker. At the same time, we see no need in inflating the manuscript length by repeating what already has been published.

It seems that the used voucher information, sampling points, and Genbank accession numbers are absent. The authors wrote ‘Genbank accession numbers of the included 1076 specimens and vouchers of specimens studied de novo are compiled in Appendix 1’. However, supplementary materials do not contain any information mentioned. That is why it is hard to say the originality of the results of the present study.

--- We checked that the voucher information and Genbank accession numbers were included in the original submission as Appendix1 and cannot explain why the reviewers could not access it. Hopefully all Supporting information it will appear in the published version.

I think the map of the collection sites would be also helpful.

--- Here we differ in our opinions with the reviewer, as our study was not aimed at the revision of distribution of particular species and hence our distribution data would necessarily be incomplete, sometimes possibly even biased. At the same time, most of the studied genera have largely overlapping ranges and do not show patterns helpful in the delimitation of newly recognized genera.

A little more about supplementary materials. I’ve found only one file with only one tree without any figure caption, that is why it is hard to say what is especially described on that. Nevertheless, in the text there are several linkages to supplementary files (e.g., Figures S1 and S2, Appendix 1), but it looks that materials are skipped on their own. Thus, it is not possible to check the propriety of the description of some parts of the text at the moment. These points need to be confirmed and all figures must be presented in the manuscript.

Again, we are unable to explain the fate of obviously submitted regular and supplementary graphical files.

In M&M part the information about indels coding is also limited. It seems that that information is important because the authors applied a different strategies of analyzing these dataset (lines 1086-1095). However, the information of how many indels, their positions in alignment, those were coded is omitted. The best practice is to upload alignments in the public data repository, e.g. TreeBASE, especially if the indels information used for the phylogeny reconstructing.

--- The information of number of sites provided by coded indels was included in the results; specifying their positions in the alignment seems to be hardly possible in the text. The alignment can be added as a supplementary file if the editor would suggest this

Please, also clarify which substitutional model was used for indel datasets.

---For indels, the uniform rates model was set (default nst setting for lset command).

Because of quite informed discussion, I would strongly recommend presenting the conclusion as a separate section. Right now, the ‘Concluding remarks’ presents as a subsection of the discussion. It seems the conclusion can be prepared based on the aforementioned subsection but more concise, or prepare de novo.

---Concluding remarks are separated now to a chapter of its own. However it still includes the general conclusions and possibilities of further development/reconsidering of the obtained data in broader (than taxonomy) context instead of repeating the key results, which are considered in the Abstract

The resolution of Figure 1 is too poor, the node support levels as well as some taxon names are not readable.

---Agreed, more detailed version of the Figure 1 is submitted for publishing

The correspondent author is not indicated and the section of the author contributions is omitted.

--- The corresponding author will probably only appear in the final version of the manuscript.

I think the English polishing would also improve the manuscript.

--- We checked again the English in the manuscript and adopted some changes. This process is kind of endless and prone to subjective perception, yet we cannot disagree on the fact that the style and wording always could be improved.

Reviewer 2 Report

The article under review is in the field of taxonomy. The aim of the work was to present a new taxonomic arrangement of species from the genera Dicranella s. l. and Aongstroemia s.l. The authors used molecular methods and conducted classical anatomical and morphological studies on selected species.

I consider the most valuable part of the work to be the description of taxa new to science, very well illustrated. The descriptions of the new nomenclature combinations are also correct, with the corresponding synonyms indicated.

Due to the fact that the studied groups of species are relatively poorly known, the work will certainly be of great interest to bryologists.

I believe that at the end of the paper, the authors should include the key to the identification of the studied species.

I have made some notes in the attached file.

In my opinion, this work fully deserves publication in the journal Plants.

Author Response

I believe that at the end of the paper, the authors should include the key to the identification of the studied species.

---We compiled diagnoses and descriptions of the newly established taxa in a way which we hope allows for their unambiguous recognition. At the same time, we deliberately did not include the key to the species identification since our manuscript is essentially not a classical taxonomic revision of genera which would deal with all described species but rather a generic revision which in individual cases identified taxa at species rank which were either undescribed or abandoned or recognized incorrectly as infraspecific taxa. In these cases, the species were properly described and their differentiation from similar taxa discussed in detail.

I have made some notes in the attached file.

---Suggestions in notes were followed

Round 2

Reviewer 1 Report

Dear Authors,

In general, the text has been corrected according the all my comments. The text looks ready to publish now. This time, the corrected version of the MS presents the supplementary materials in full. I still wish to point out that for a high level of publication (e.g., Q1 journals), it is good practice to provide a link to DNA alignments used for phylogeny reconstruction uploaded to open data repositories so that the data can be checked by other researchers. Such a repository is a TreeBASE or others. I also suggest renaming 'Concluding remarks' paragraph to 'Conclusions' in accordance with the Instructions for Authors.

Author Response

Thank You for the helpful suggestion, indeed publishing the dataset in the public repository would probably better than doing this via supplementary info in the paper page. So, here is a reviewer link to the .nex file in the TreeBASE

http://purl.org/phylo/treebase/phylows/study/TB2:S30163?x-access-code=4e11e8c1da723d22d30bcf460d0e84f8&format=html